# Out of Spuriousity: Improving Robustness to Spurious Correlations without Group Annotations

**Phuong Quynh Le**                                       *phuong.le@uni-marburg.de*
*Marburg University*

**Jörg Schlötterer**                                *joerg.schloetterer@uni-marburg.de*
*Marburg University & University of Mannheim*

**Christin Seifert**                                   *christin.seifert@uni-marburg.de*
*Marburg University*

**Reviewed on OpenReview:** *https://openreview.net/forum?id=EEeVYfXor5*

## Abstract

Machine learning models are known to learn spurious correlations, i.e., features that have strong correlations with class labels but no causal relationship. Relying on these correlations leads to poor performance in data groups that do not contain these correlations, and poor generalization. Approaches to mitigate spurious correlations either rely on the availability of group annotations or require access to different model checkpoints to approximate these group annotations. We propose PRUSC, a method for extracting a spurious-free subnetwork from a dense network. PRUSC does not require prior knowledge of the spurious correlations and is able to mitigate the effect of multiple spurious attributes. Specifically, we observe that Empirical Risk Minimization (ERM) training leads to clusters in representation space that are induced by spurious correlations. We then define a supervised contrastive loss to extract a subnetwork that distorts such clusters, forcing the model to learn only class-specific clusters, rather than attribute-class specific clusters. Our method outperforms all annotation-free methods, achieves worst-group accuracy competitive with methods that require annotations and can mitigate the effect of multiple spurious correlations. Our results show that in a fully trained dense network, there exists a subnetwork that uses only invariant features in classification tasks, thereby eliminating the influence of spurious features.

## 1 Introduction

Deep neural networks tend to learn spurious correlations or shortcuts, i.e., misleading heuristics that work for the majority of training samples, but do not always hold (Sagawa et al., 2020a). Spurious attributes or features are features that have no causal relationship with class labels. The existence of spurious features in a data set itself is not harmful, as long as a model does not rely on them for the classification. However, spurious features are often easy to learn, and if there is a strong correlation with class labels, i.e., a spurious correlation, models tend to rely on these spurious correlations as shortcuts, because learning these correlations is sufficient to minimize the overall loss (simplicity bias) (Shah et al., 2020). This means, that in the presence of spurious correlations, deep neural networks under-learn invariant information and then lose their ability to generalize and perform poorly on data that does not contain the spurious correlations.[1] Spurious features occur naturally, and some of them, such as image background, can be easily detected, while others, such as

---

[1]Shortcut learning can also occur on (subsets of) non-spurious features that do not fully describe the relationship between features and class labels. Ideally, we aim for robust models that rely on invariant features, i.e., they do not rely on a particular distribution of the training data.

texture (Geirhos et al., 2018), superficial statistics (Wang et al., 2019), or frequency bias (Wang et al., 2022; 2023), require more effort. An example of a natural spurious correlation of the background is the *cows* and *camels* classification task, where the background *desert* is spuriously correlated with the class *camels* and *grass* is positively correlated with *cows* (Beery et al., 2018). These features seem to be easy to learn for deep networks, but relying on them makes networks perform poorly in *minority groups* such as *camels* on *grass.*

To improve the performance for minority groups, mitigation strategies use group label information, obtained either by manual labeling or by approximation using misclassification information. Ground-truth group labels have been used during training to minimize the worst-group loss (Sagawa et al., 2020a) or to retrain only the last layer (Kirichenko et al., 2023). However, obtaining group labels either requires prior knowledge of the spurious attributes and a time-consuming annotation effort, or may be impossible for certain types of attributes. For example, neural networks tend to rely on low frequency components for regression tasks (Rahaman et al., 2019). Wang et al. (2022; 2023) showed that classification models also rely on frequency (both texture-based and shape-based). However, frequency attributes are only implicit and not visible to the human eye, unlike explicit spurious attributes, such as background (e.g., grass or desert). A common proxy estimate for missing group labels is to use misclassified examples from the training data (i.e., hard cases), based on the assumption that models that are influenced by spurious correlations fail to classify instances where the spurious correlation is absent (Liu et al., 2015; Nam et al., 2020; Zhang et al., 2022; Park et al., 2023). However, this assumption is not guaranteed to hold. Furthermore, in the case of zero training loss, where the model fits the training data perfectly, hard cases are not available. Careful control of early stopping is then required to obtain a sufficient number of hard cases without degrading the Empirical Risk Minimization (ERM) training. Consequently, these methods also require access to full training information and are not applicable to fully trained models.

In this paper, we propose Pruning Spurious Correlations (PRUSC), a method for extracting a spurious-free subnetwork from a fully trained dense network that requires no prior knowledge of the spurious correlations, i.e., no ground-truth group annotations. PRUSC defines potential spurious groups based on the representation of instances in latent space. Based on the observation that instances with the same spurious attributes lie close together in representation space, we can identify groups of instances with the same spurious feature. We apply clustering to the representations of the penultimate layer and introduce a novel application of supervised contrastive loss to train a subnetwork such that clusters (likely) induced by spurious attributes are *distorted.* By distorting the clusters induced by the spurious feature, the subnetwork cannot rely on learning and grouping samples based on similar spurious attributes and is, therefore less influenced by spurious correlations. In summary, the contributions of this paper are:

- We present PRUSC, an approach to mitigate the effects of spurious correlations. Our method relaxes the assumptions of previous work: PRUSC does not require prior knowledge of spurious correlations, and it does not require any group annotations in either the training or the validation set.
- We show how to extract a subnetwork optimized for unlearning data manifolds induced by spurious attributes. We base our solution on contrastive learning, and introduce an effective way to sample contrastive batches based on representation clustering.
- We evaluate PRUSC on benchmark datasets and show that it has superior worst-group accuracy to annotation-free approaches and is competitive with approaches that require annotations.
- We show that PRUSC is robust to multiple spurious correlations and achieves the largest improvement over all spurious attributes.

We evaluate our approach on the CelebA dataset (Liu et al., 2015) with potentially multiple spurious attributes, the ISIC skin cancer dataset (Codella et al., 2019), a realistic collection of images used for melanoma detection, and the artificial benchmark Waterbirds (Sagawa et al., 2020a). Our method outperforms all approaches that do not require group annotations across all benchmarks and has the highest overall score on 2 out of 3 benchmarks. Our code is available at: `https://github.com/aix-group/prusc`.

## 2 Related Work

In this section, we review related work on mitigating spurious correlation. Approaches to mitigate spurious correlations can be broadly categorized based on assumptions: methods that require prior knowledge of

spurious correlations, such as group labels or human annotations, and methods that do not require prior knowledge.

With prior knowledge of spurious correlations, training regimes can directly mitigate undesired attribute influences or relations in the data and consequently yield high accuracy, both on average and in the minority group. Data-centric approaches such as UV-DRO (Srivastava et al., 2020) and LISA (Yao et al., 2022) assume full information about the type of the spurious correlations on all instances and augment the training data with additional samples to eliminate the spurious correlation. GroupDRO (Sagawa et al., 2020a) instead uses the existing group labels directly to minimize the loss for the worst-group performance. Requiring only a small subset of group labeled training data, Barack (Sohoni et al., 2022) predicts other instances' groups and DFR (Kirichenko et al., 2023; Izmailov et al., 2022) retrains part of the model with a small, spurious-free dataset. Group annotation or *human-in-the-loop* annotations is, however, costly and difficult to achieve in practice. In particular complex spurious attributes, such as as fluency bias (Wang et al., 2023) or texture and shape bias (Geirhos et al., 2018) remain challenging, as they are difficult for humans to detect.

Assuming no group annotations, Sohoni et al. (2020); Seo et al. (2022) and Creager et al. (2021) focus on automatically inferring appropriate groups before training a robust model (e.g., GroupDRO) with pseudo-group labels. Several methods follow a two-stage training approach, where the first stage is commonly used to identify hard cases from ERM training. These hard cases are instances that the ERM model fails to classify, presumably because ERM relies on spurious correlations, with the hard-to-classify cases not exhibiting the spurious correlation, i.e., bias-conflicting samples. Not utilizing the misclassified cases, SPARE (Yang et al., 2024) infers groups labels by clustering model outputs in early epochs, then applies importance sampling to upsample smaller clusters (minority groups) and downsample larger clusters (majority groups) during the remaining epochs. Approaches differ in the second stage: Liu et al. (2021); Nam et al. (2020) upweight hard cases in an additional training run, while Yaghoobzadeh et al. (2021) exclusively fine-tune hard cases. CnC (Zhang et al., 2022) and DCWP (Park et al., 2023) utilize the set of hard cases for contrastive learning to define contrastive batches and retrain the whole network (CnC) or to extract a subnetwork (DCWP) with contrastive loss. Different from the two-stage approaches, DeDiER (Tiwari et al., 2024) uses knowledge distillation to find overconfident predictions in early network layers $d$, and then uses this signal to weight the distillation loss on a per-instance level. SELF (LaBonte et al., 2024) finds a small set of hard cases by measuring the degree of disagreement in predictions between an ERM and an early-stop ERM model, and then applies last-layer fine-tuning with this set. The effectiveness of methods that rely on ERM misclassifications for hard cases depends on both the quantity and the quality of the misclassifications. In cases where ERM models almost perfectly fit the training data (training accuracy close to 100%), there are not enough hard cases (i.e., misclassifications) and these methods fail. To obtain a sufficiently large set of hard cases, methods either use a carefully controlled early stopping criterion (Liu et al., 2021) or classify based on earlier layers in the networks, assuming that these layers still contain information about misclassified samples, likely originate from learned spurious correlations (DeDiER (Tiwari et al., 2024)). Therefore, methods that rely on misclassification cannot directly mitigate spurious correlations of a trained ERM model unless trainers have access to the entire training process of that model.

## 3 Problem Setting and Example

In this section, we describe spurious correlations in machine learning more formally (Sec.3.1), and show the core idea of our method PruSC for mitigating spurious correlations in a simple example (Sec.3.2).

### 3.1 Problem Setting

Suppose a data set $\mathcal{D}$ consists of data where each sample has some attributes of the input $x$ that are correlated with the target labels $y \in \mathcal{Y}$. With prior knowledge of *spurious attributes $a \in \mathcal{A}$*, we can partition the data set $\mathcal{D}$ into groups $g \in \mathcal{G}$ according to $\mathcal{G} = \mathcal{A} \times \mathcal{Y}$, i.e., each group $g$ is defined by the combination of the label and the corresponding spurious attribute (Sagawa et al., 2020b).

A model trained with Empirical Risk Minimization (ERM) on the class labels may be able to separate the classes well but rely on spurious attributes to do so, resulting in a mismatch between the intended and the

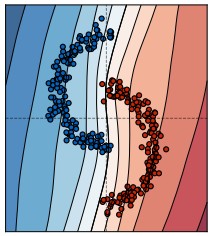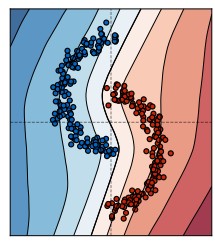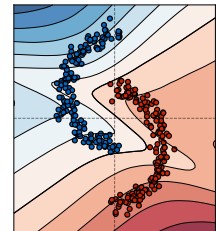

Figure 1: Simple network on the two moons dataset. Decision boundaries (black lines) depend on model capacity and loss. Left: Standard ERM with cross-entropy loss with a nearly linear decision boundary shows a strong dependence on the x-coordinate (nearly linear vertical decision boundary). Center: Pruned network (masking) using only 50% of the weights at test time shows less dependence on the x-coordinate. Right: Masking, and using our contrastive loss (cf. Sec. 5.2.1) results in non-linear decision boundaries with large margins.

model-learned solution (Geirhos et al., 2020). Spurious attributes typically appear in the majority of the dataset. Thus, by using these attributes, an ERM model can minimize the training loss for the expected average population, but still show high errors on the minority group that lacks these spurious attributes. Furthermore, in cases where the spurious attributes are easy to learn (e.g., background instead of the object), ERM prioritizes learning those attributes over more complex ones and then loses the ability to generalize. Our goal is to mitigate the learning of spurious correlations by addressing both statistical (highly unbalanced group distributions) and geometric (the tendency to learn simpler features) issues.

### 3.2 Motivating Example

To illustrate the problem and motivate our approach, we consider a simple 2-D classification task using the two moons dataset (Pezeshki et al., 2021), with one attribute that is strongly correlated with target labels, causing neural networks to under-learn other invariant attributes. The two moons dataset $\mathcal{D}$ consists of input pairs $(x, y)$, where $x = (x_1, x_2) \in \mathbb{R}^2$ and $y \in \{0, 1\}$ (cf. Fig. 1). The dataset contains a strong relation between $x_1$ and the corresponding label $y$. We train three different models on this dataset, showing the effect of i) smaller networks and ii) our contrastive loss formulation (Sec. 5.2.1). A simple feed-forward network (5 ReLu layers, 500 hidden units per layer, 100 epochs, cross-entropy loss) can separate the two classes very well (Fig. 1, left). The decision boundary is nearly linear because the cross-entropy loss optimizes for class separation and for the model it is sufficient to rely on the correlation between the x-axis values of $x$ and $y$ to minimize the loss. We train the same network architecture with random masks (50% of weights in each layer) and use the same fraction of random weights for testing, i.e., we randomly prune the network to 50% of its weights. We observe curved decision boundaries (Fig. 1, center), suggesting that a smaller network within a dense network is more robust to invariant features. Combining masked training and our contrastive loss (Sec. 5.2.1, we obtain a curved decision boundary with a large margin (Fig. 1, right), suggesting that the model uses information from both the x- and y-coordinates.

The above example strengthens **two hypotheses** for mitigating spurious correlations. First, **network capacity** seems to have an influence. Both, the dense network and the smaller subnetwork with randomly pruned connections perform well on the dataset, but the pruned network has better decision boundaries, i.e., depends less on the spurious feature. However, this does not suggest to train a sparse architecture from scratch, because small architectures are difficult to train, whereas larger models have a higher ability to learn information and improve performance beyond the bias-variance regime (Nakkiran et al., 2021; Frankle & Carbin, 2018). Instead, it suggests to first train a large network with sufficient capacity to encode all relevant information, and then extract a subnetwork after training that does not rely on spurious correlations. The post-training approach is also consistent with our realistic scenario that the existence of spurious correlations becomes apparent only after training (and possibly later in the lifetime of a model). Thus, extracting a subnetwork without spurious correlations is less expensive than retraining the whole model from scratch. This observation is our main motivation. Second, our **contrastive learning** scheme improves decision boundaries

by learning more information from the training data than just the simplicity bias features. Our batches of contrastive instances serve as a guide for the models to know which feature relations to focus less on.

## 4 Background: Subnetwork Extraction

In this section, we outline the approach for the extraction of subnetworks (Csordás et al., 2020), i.e., subnetworks responsible for a specific task. The extraction method is also utilized by Zhang et al. (2021) to solve out-of-distribution generalization by probing subnetworks serving different functional subtasks from a pre-trained model. Additionally, Park et al. (2023) use this approach to train a subnetwork that more robust to a specific set of hard cases, therefore, mitigating the effects of spurious correlations in learning. In generally, we aim to extract a sparsely connected subnetwork from a dense network with a binary mask indicating which weights to include (the subnetwork) and which to discard.

**Masking Models.** Given a trained neural network $f$, for layer $l$ in $L$ hidden layers of a neural network $f$, we introduce a learnable parameter $\mathbf{\Pi} = (\pi^1, \pi^2, ..., \pi^L)$ corresponding to the weights of each layer $\mathbf{W} = (\mathbf{w^1}, \mathbf{w^2}, ..., \mathbf{w^L})$. Each element $\pi_{j,k}^l$ acts as a logit indicating the probability of keeping the corresponding weight $w_{j,k}^l$. The weight $w_{j,k}^l$ connects the $j^{th}$−neuron from the $(l-1)^{th}$−layer to the $k^{th}$−neuron from $l^{th}$−layer, we write each weight and its corresponding logit as $w_i$ and $\pi_i$ for short. During training, network parameters in hidden layers become $(\mathbf{w^1} \odot \pi^1, \mathbf{w^2} \odot \pi^2, ..., \mathbf{w^L} \odot \pi^L)$.

**The Mask Training.** To train the binary mask, we freeze the model weights $\mathbf{W} = (\mathbf{w^1}, \mathbf{w^2}, ..., \mathbf{w^L})$ and train only the added parameters $\mathbf{\Pi} = (\pi^1, \pi^2, ..., \pi^L)$ with each $\pi_i$ equal to 0.9 initially, i.e., high keep probability. Modular loss $\mathrm{L_{mod}}$ (Csordás et al., 2020) aims to extract a set of sparse weights (the subnetwork) that retain the performance of the original dense network on the classification task.

$$\mathrm{L_{mod}} = \mathrm{L_{CE}} + \alpha \sum_i \pi_i, \tag{1}$$

where $\mathrm{L_{CE}}$ is the cross-entropy loss, $\sum_i \pi_i$ is sparse regularization, which keeps a logit $\pi_i$ small unless it is needed for the task and $\alpha$ is responsible for the strength of the sparse regularization.

The Gumbel-Sigmoid trick (Jang et al., 2017) is applied during training to the logit

$$s_i = \sigma((\pi_i - \log(\log U_1 / \log U_2)/\tau) \text{ with } U_1, U_2 \sim \mathrm{U}(0,1), \tag{2}$$

where $\tau$ is the temperature and $\sigma(x)$ is the sigmoid function. We obtain the binary mask by

$$m_i = [\mathbb{1}_{s_i > \gamma} - s_i]_{stop} + s_i, \tag{3}$$

where the threshold $\gamma$ is set to 0.5 by default, $\mathbb{1}$ is the indicator function and $[.]_{stop}$ is the stop gradient operator. Accordingly, the binary mask $m_i = 1$, if the probability $s_i$ to keep this weight exceeds 0.5 and $m_i = 0$ otherwise. The final parameters of the resulting subnetwork are defined as $\mathbf{W'} = (\mathbf{w^1} \odot \mathbf{m^1}, \mathbf{w^2} \odot \mathbf{m^2}, ..., \mathbf{w^L} \odot \mathbf{m^L})$.

## 5 Our Approach

We hypothesize that instances with the same spurious attribute are nearby in representation space, and that spurious attributes induce clusters (cf. illustration in Fig.3 A). We verify this hypothesis for multiple attributes on the CelebA dataset (Sec. 5.1). Our approach centers around two key ideas: First, we identify clusters in the latent space and move instances with the same spurious attributes away from each other and samples of the same class closer to each other. Second, we reduce the representation capacity of the network to a subnetwork that is less prone to spurious correlations. Specifically, we learn a task-oriented subnetwork (Sec. 5.2) with a task-specific contrastive loss (Sec. 5.2.1) that optimizes the representation space. We describe the specific data selection and sampling procedure for negative and positive samples in the contrastive loss in Sec. 5.2.2. Finally, we present the overall training procedure in Sec. 5.3.

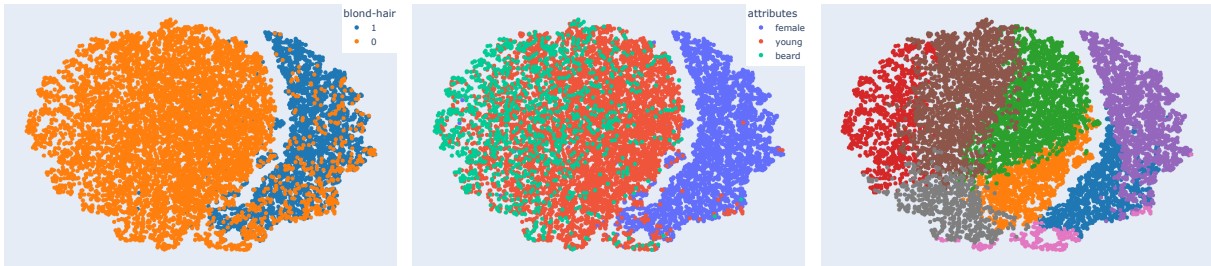

Figure 2: Embedding space (t-SNE) of ERM on CelebA for predicting hair color. Colors represent class labels (left), attributes (center), and k-means cluster labels (right). Spurious attributes are strongly correlated with class labels (e.g., female – blond hair) and sub-manifolds are defined by spurious attributes within a class (non-blond, beard and young). $k-$means tends to cluster based on attributes with high purity.

## 5.1 Analysis of ERM Representation Space for Multiple Spurious Attributes

In the presence of spurious attributes, ERM tends to learn them as predictive features yielding decision boundaries that are aligned with the spurious feature (as exemplified on the two moons dataset in Fig. 1). We analyze the behavior of ERM in the presence of multiple spurious features on the CelebA dataset. We use an ERM model for predicting hair color (blond or not blond) having a training accuracy of 95%.

To analyze the feature space, we select three additional attributes (gender, age, presence of a beard) as potential spurious attributes. For a brief description of the CelebA dataset see Sec. 6.1 and Appendix A.1 for detailed information. Fig. 2 shows the t-SNE (Van der Maaten & Hinton, 2008) projection of a random subset of CelebA training data representations from the penultimate layer[2]. The two classes are well separated in representation space (Fig. 2, left). Interestingly, there is a strong correlation between spurious attributes and classes in the representation space. Female correlates with blond hair, whereas the cluster for non-blond hair (orange cluster, left in Fig. 2) corresponds to young people and people with beard. Furthermore, within a class (non-blond hair), the two attributes partition the data into manifolds, even though the attribute labels are not used during training. Finally, unsupervised k-means clustering in representation space tends to cluster by attributes with high purity (Fig. 2, right). This means that when optimizing for class separation, instances from the same attribute group $a_i \in \mathcal{A}$ (e.g., persons with beard, young persons) lie close together in representation space, even though the group labels are not used during training.

Tab. 1 shows the purity of classes (blond or non-blond hair) and spurious attributes (male or female) in clusters obtained from $k$-means clustering on representations of the CelebA training set. Each cluster is assigned a label based on the most frequent class/ spurious attribute. Purity is calculated as the number of correctly matched attribute and cluster labels divided by the total number of instances in that cluster. The table demonstrates that $k$-means clustering effectively groups samples by class label, but achieves even higher purity for spurious attributes. Fig. 2 (right) demonstrates $k$-means clustering ($k = 8$) well defines clusters based on not only one but multiple spurious attributes. The critical case is when the model can potentially separate both, classes and attribute groups, i.e., when the model has high predictive performance **and** the latent space shows clear clusters for spurious

Table 1: Purity measurement according to class labels ($\mathcal{Y}$-purity) and spurious attributes ($\mathcal{A}$-purity) of unsupervised $k$-means clustering with different values of $k$.

| $k$ | $\mathcal{Y}$-purity | $\mathcal{A}$-purity |
|---|---|---|
| 4 | 95.55% | 96.48% |
| 8 | 97.01% | 97.32% |
| 16 | 97.00% | 97.80% |

attributes. Such a model will i) make predictions based on spurious attributes and ii) not generalize well to samples with attribute combinations that do not align with the spurious correlations.

Yang et al. (2024) investigate and provide a theoretical proof using a two-layer fully connected neural network trained on the CMNIST dataset (which has a spurious correlation between each class and a specific color) to mimic the early learning process of neural networks. The authors show that each group (defined by a label

---

[2]We discard samples with more than one of the selected attributes for easier interpretation.

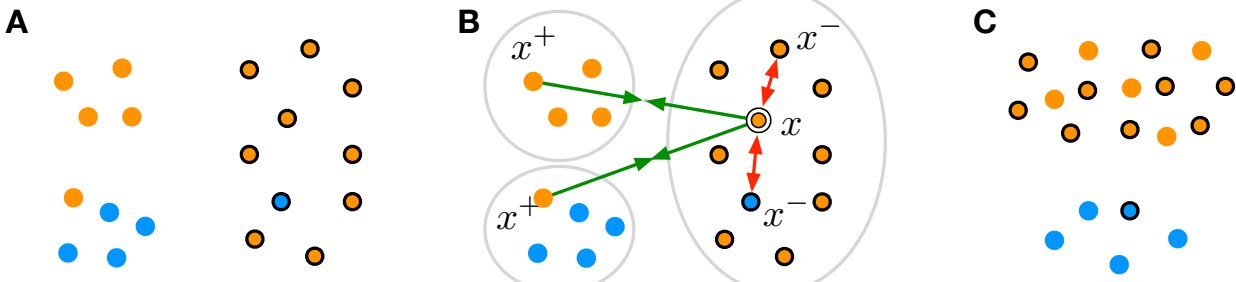

Figure 3: Assumption and overall idea. **A.** Instances from the same class (same colour) lie in different clusters apart in feature space. Instances with the spurious feature (solid border) are nearby in feature space, i.e., the spurious feature induces clusters and primarily defines the shape of the data manifold. **B.** Contrastive loss: For an anchor $x$, instances from the same class but different clusters constitute positive samples $x^+$, instances from the same cluster negative samples $x^-$. Positive samples are moved closer to the anchor, negative samples away from the anchor. In each iteration multiple samples act as anchors (with their positive and negative samples). **C.** Goal and effect of our contrastive loss. The spurious feature does not define data manifolds and cannot be used as discriminative feature between classes. **Note:** Our approach does not assume (and need) knowledge about the spurious feature(s).

and its correlated spurious feature) can be separated *early in training* based on the model's outputs. Our observations empirically show that even with a complex neural network, which is assumed to learn simple functions early in training and gradually build complexity over time, the representation space can still be separated by spurious features *after training* once spurious correlations are learned.

Based on these observations, our goal is to ensure that spurious attributes do not define data manifolds and cannot be used as discriminative attributes.

## 5.2 Task-oriented Subnetworks

To ensure that spurious attributes do not define data manifolds and cannot be used as discriminative attributes, we introduce a task-oriented subnetwork, and a contrastive loss that optimizes the representation space. More specifically, we extend the approach for extracting subnetworks (cf. Sec. 4) by incorporating a *task-specific loss* function to prevent the network from learning a grouping of samples based on spurious attributes. The loss for our task-oriented subnetwork is defined as

$$L = L_{mod} + \beta L_{task}, \tag{4}$$

where $L_{mod}$ is the modular loss (cf. Sec. 4), $L_{task}$ the task-specific loss, and $\beta$ is a hyper-parameter to balance the loss terms.

The choice of $L_{task}$ in combination with a mask-training dataset $\mathcal{D}_{task}$ defines the specific task the subnetwork is optimized for. For example, Csordás et al. (2020) finds a subnetwork responsible for a particular class ($\mathcal{D}_{task}$ contains examples of this class, $L_{task} = 0$), and Park et al. (2023) identify a subnetwork that is more robust to a spurious feature ($\mathcal{D}_{task}$ contains instances misclassified by ERM, and $L_{task}$ is a debiasing term). Details on these methods can be found in Appendix A.2.

We detail our choice of $L_{task}$ and $\mathcal{D}_{task}$ in Sec. 5.2.1 and 5.2.2, respectively. Fig. 3 illustrates the training procedure of our task-oriented subnetwork. Based on the assumption that instances of the same class (same color) lie in different clusters, and that spurious attributes induce clusters, we define a contrastive loss $L_{task}$, which moves samples of the same class in different clusters closer together in representation space, and pushes samples of the same cluster apart.

### 5.2.1 Contrastive Learning

We use a variant of contrastive loss (Khosla et al., 2020) to mitigate the tendency of ERM to form clusters based on spurious attributes. Different from the original work, our contrastive batch is defined not only by the class labels, but also by the location of the samples in representation space. Fig. 3 outlines the idea.

For each contrastive batch we randomly sample a set of *anchors* and for each anchor a corresponding set of positive and negative samples. We define positive samples as samples of the same class as the anchor, but from a different cluster than the anchor. We define negative samples as samples from the same cluster (independent of their class label). For anchor $(x, y)$ belonging to cluster $c \subset \mathcal{C}$, the positive set of size P is $\{(x^+, y^+) \mid (x^+, y^+) \in \mathcal{C} \setminus c \wedge y^+ = y\}$ and the negative set of size N is $\{(x^-, y^-) \mid (x^-, y^-) \in c \setminus (x, y)\}$. The contrastive loss term defined with respect to the *anchor* $(x, y)$ is

$$l_{\text{con}}(x) = -\sum_{i=1}^{P} \log \frac{\exp(z^\top z_i^+/\tau)}{\sum_P \exp(z^\top z_p^+/\tau) + \sum_N \exp(z^\top z_n^-/\tau)}, \tag{5}$$

where $\tau > 0$ is a scalar temperature hyper-parameter, $z, z^+, z^-$ are the representations obtained from the penultimate layer of the corresponding inputs $x$, $x^+$ and $x^-$ from *anchor*, *positives* and *negatives* respectively. We follow the finding of (Khosla et al., 2020) that locating the summation *outside* the log improves performance over summation *inside* the log. This loss forces the representations of samples of the same class across clusters to become more similar and more similar to the *anchor*, while samples representing potentially spurious attributes are pushed away from the *anchor*.

### 5.2.2 Data

PRUSC does not require prior knowledge of spurious attributes and is applicable in the presence of multiple spurious attributes. We do not assume that group annotations are available but automatically infer groups to construct the de-biasing dataset $\mathcal{D}_{\text{task}}$. Based on the assumption that the representation space of ERM has regions corresponding to spurious attributes (cf. Sec. 5.1, and Fig. 2), we use unsupervised clustering to construct the spurious groups for our dataset $\mathcal{D}_{\text{task}}$.

Given an ERM model, we calculate the embeddings of the training data $\mathcal{D}$ and apply $k$-means clustering on the embeddings. Each sample in the training set $\mathcal{D}$ is assigned to a cluster $c$, $(x, y, c) \in \mathcal{D} = \mathcal{X} \times \mathcal{Y} \times \mathcal{C}$, with input domain $\mathcal{X}$, set of class labels $\mathcal{Y}$, and set of clusters $\mathcal{C}$. We define the number of classes in the classification task as $|\mathcal{Y}| = K$ and the number of clusters $|\mathcal{C}| = k$.

We distinguish clusters based on their class purity. A cluster c is called $i-$dominant for class $i$ if at least 90% of its samples belong to class $i$, i.e., a cluster is $i-$dominant, if $\exists i \in \mathcal{Y}$ such that $\frac{|\{(x,y,c) \in c|y=i\}|}{|c|} \geq 0.9$. Otherwise, c is *neutral*. A cluster must fall into one of the $K + 1$ categories that are either neutral or $i-$dominant for some $i \in \mathcal{Y}$ and multiple different clusters may be $i-$dominant for the same class $i$. $C^i$ is the set of $i-$dominant clusters (of the same class $i$) and $C_N$ is the set of neutral clusters.

Based on the observation in Section 5.1 that cluster purity is higher w.r.t. attributes than w.r.t. to classes, we derive that minority instances in an $i-$dominant cluster are typically hard cases. Hard cases contain a spurious attribute, but not the spurious correlation, i.e., are assigned to a class other than the spuriously correlated one. Accordingly, the set of hard cases $h_i$ obtained from the representation space contains samples with labels $i$ assigned to clusters dominated by a different class:

$$h_i = \{(x, y, c) \mid c \in C^{j \neq i} \wedge y = i\} \tag{6}$$

To construct a class-balanced dataset $\mathcal{D}_{\text{task}}$ for subnetwork training, we determine a fixed number $p$ of samples for each class. We draw $p$ samples for class $i$ as follows: First, we take all samples from the hard cases set $h_i$. Second, we draw the remaining samples for class $i$ uniformly from all $i-$dominant and neutral clusters. That is, from each cluster in $C^i$ and $C_N$, we draw an equal number of $\frac{p-|h_i|}{|C^i|+|C_N|}$ samples[3] having class label $i$. We obtain a subset of the training data $\mathcal{D}_{\text{task}}$ that is class balanced with $p$ samples for each class.

---

[3] In the exceptional case $|h_i| + |C| = 0$ (no neutral cluster exists and all clusters are dominant for some class other than $i$) we only use $h_i$, which may result in less than $p$ samples. We never encountered such a situation in our experiments.

By sub-sampling $\mathcal{D}_{\text{task}}$, we intend to construct a subset from the training data without spurious correlations. In order to mine hard cases without prior knowledge of spurious correlations or access to ERM misclassifications from pre-training, we sample minority cases from $i-$ dominant clusters. These instances are supposed to be anti-correlated. In addition, we draw remaining samples for class $i$ uniformly from all $i-$dominant and neutral clusters to construct a balanced dataset, i.e., without a spurious correlation. Drawing an equal number of samples per class results in a balancing, both along (spurious) features and classes. We also use that balanced dataset $\mathcal{D}_{\text{task}}$ for fine-tuning the extracted subnetwork after pruning, in order to avoid re-learning the spurious correlations. The ablation results in Sec. 7.1 confirm that fine-tuning with $\mathcal{D}_{\text{task}}$ is an effective performance improvement.

## 5.3 Overall Training Procedure

Given a trained ERM neural network $f$ on the dataset $\mathcal{D}$, we perform three steps sequentially.

**Representation Clustering and Sub-sampling.** First, we extract the representations learned by the ERM model as the output of the last hidden layer of $f$: $f_{emb}$. We apply $k-$means to $f_{emb}(x)$ with a pre-defined $k$.[4] We determine cluster labels ($i-$dominant and neutral) and construct the subnetwork mask-training dataset $\mathcal{D}_{\text{task}}$ as described in Sec. 5.2.2. In our experiments, we set $p$, the number of samples for each class such that $\mathcal{D}_{\text{task}}$ typically contains 10% of the samples of $\mathcal{D}$.

**Binary Mask Training.** We freeze all weights $\mathbf{W}$ of $f$ and train only the mask $\mathbf{\Pi}$ with the mask-training dataset $\mathcal{D}_{\text{task}}$ (cf. Sec. 4). To construct contrastive batches, we randomly sample multiple *anchors $x$* from different clusters along with their corresponding positives $x^+$ and negatives $x^-$. We accumulate the per-batch constrastive loss by summing over $\text{L}_{\text{con}}$ of all *anchors* in the batch. The mask $\mathbf{\Pi} = (\pi^{\mathbf{1}}, \pi^{\mathbf{2}}, ..., \pi^{\mathbf{L}})$ is updated through the final loss term

$$\text{L} = \text{L}_{\text{CE}} + \alpha \sum_i \pi_i + \beta \text{L}_{\text{con}}, \tag{7}$$

where cross-entropy loss $\text{L}_{\text{CE}}$ and sparse regularization $\sum_i \pi_i$ correspond to $\text{L}_{\text{mod}}$ for extracting a sparse subnetwork, and $\text{L}_{\text{con}}$ is the contrastive loss to mitigate the formation of spurious attribute clusters.

**Fine-tuning.** After binarization of the trainable mask $\mathbf{\Pi}$, the resulting binary mask $\mathbf{m}$ defines the parameters of the subnetwork as $\mathbf{W}' = (\mathbf{w^1} \odot \mathbf{m^1}, \mathbf{w^2} \odot \mathbf{m^2}, ..., \mathbf{w^L} \odot \mathbf{m^L})$. We fine-tune the extracted subnetwork on the dataset $\mathcal{D}_{\text{task}}$ with cross-entropy loss for a few epochs.

# 6 Experiments

We evaluate the effectiveness of PRuSC on three datasets (CelebA, Waterbirds, and ISIC) in comparison to state-of-the-art methods (Sec. 6.4). We further examine the impact of PRuSC in the presence of multiple potential spurious correlations (Sec. 6.5).

## 6.1 Datasets

**CelebA.** Following previous work, our classification target is hair color (blond or non-blond hair) on the CelebA dataset (Liu et al., 2015), where the potential spurious attribute is gender (male or female). Majority groups are blond females and non-blond males with proportions of 44% and 41% of the data. The minority groups are non-blond females (14%) and blond males (1%). We extend the common setup to a multi-spurious dataset by incorporating additional attributes (no beard, young, heavy makeup, wearing lipstick, pale skin) that also pose potential spurious correlations based on their heavily imbalanced distributions.

---

[4]We use $k = 8$ for all experiments and show that the number of clusters not a critical hyper-parameter for our approach, as long as it is large enough in section 7.2

**Waterbirds.**  Waterbirds was introduced by Sagawa et al. (2020a) as a standard spurious correlation benchmark. This artificially constructed dataset pastes bird segmentations from the CUB dataset (Wah et al., 2011) onto backgrounds from the Places dataset (Zhou et al., 2018). The birds labeled as either waterbirds or landbirds are placed on backgrounds of either water or land. The task is to classify the types of birds: $\mathcal{Y}$ = {waterbird, landbird} and pasting 95% waterbirds on a water background and 95% landbirds on a land background results in spurious correlations in the standard training set since the models tend to rely on the background instead of the actual bird characteristics. Accordingly, the spurious attributes are $\mathcal{A}$ = {water background, land background}.

**ISIC.**  ISIC Skin is a real-world medical dataset provided by the International Skin Imaging Collaboration ISIC (Codella et al., 2019). The objective is to distinguish benign (non-cancerous) cases from malignant (cancerous) cases. Using the source code of Rieger et al. (2020), we retrieved $20,394$ images for the two classes benign ($17,881$) and malignant ($2,513$) from the ISIC Archive[5]. Nearly half of the benign cases ($8,349$) have colored patches attached to patients' skin, whereas no malignant case contains such patches. In this realistic dataset, a the group "malignant with patches" is missing from the training set. To create this missing group during the validation procedure, we programmatically add colored patches to a subset of malignant cases outside the area of the lesion, following Nauta et al. (2021).

Details about the distributions of spurious attributes in each dataset are provided in Appendix A.1.

## 6.2 Baselines

Empirical Risk Minimization (ERM) represents conventional training without any procedures to improve worst-group accuracy.

We compare PruSC with two approaches that require group annotations: GroupDRO (Sagawa et al., 2020a) requires knowledge about spurious attributes and group annotations to train from scratch, while DFR (Kirichenko et al., 2023) requires a group-balanced subset for its retraining phase. To meet this requirement in the ISIC dataset, we create artificial samples for the missing group (malignant with patches) according to Nauta et al. (2021). We use these artificial samples during the training of GroupDRO and DFR. To analyze the importance of a group-balanced subset in DFR, we implement DFR$_{class}$, which is a version of DFR that uses our automatically inferred class-balanced subset $\mathcal{D}_{task}$ from clustering instead of annotated group-balanced data.

In addition, we compare to methods with relaxed annotation requirements. CnC (Zhang et al., 2022) and DCWP (Park et al., 2023) work with ERM misclassified cases instead of group annotated samples. These methods sometimes necessitate retraining the ERM with early stopping criteria to obtain a sufficiently large set of misclassified samples. SPARE (Yang et al., 2024) shows that groups in $\mathcal{G}$ can be separated early in ERM training if the spurious correlations are strong enough. It infers groups from early training epochs and from these inferred groups samples group-balanced mini batches during training of the remaining epochs. DeDiER (Tiwari et al., 2024), is an early readout mechanism with a distillation model. This method identifies misclassified cases from early layers of the network by design. However, both SPARE and DeDiER require group information during validation to tune the hyper-parameters and determine the early epoch or layer from which to take the readout. Early-stop disagreement SELF (LaBonte et al., 2024) fine-tunes the last layer weights with a small held-out dataset chosen by maximizing the disagreement between ERM model and an early-stop ERM model. SELF requires neither group annotations nor class labels for the held-out dataset, however, it does require group annotation during hyper-parameters tuning. In contrast to the above methods, PruSC does not require any group annotations, neither for (re-)training, nor for fine-tuning hyper-parameters (cf. annotation requirements in the first column of Tab. 2).

DCWP, DFR, DFR$_{class}$ and PruSC, first require an ERM model that is affected by spurious correlations before taking action to mitigate those spurious correlations. For a fair comparison, we use the same ERM model, which is the ERM baseline reported in Tab. 2 for all methods. Since the training accuracy of this baseline ERM is about 99%, DCWP needs access to the full training process of the ERM model and uses earlier model checkpoints to obtain a sufficient amount of misclassified samples.

---

[5]https://www.isic-archive.com/

Table 2: Performance overview, comparing methods with different assumptions on availability of annotations (Annot.) of the spurious features in training (Tr) and/or validation set (Val). ** denotes results reported in respective papers, n.a. denotes that results are not available in the original paper. * denotes the presence of an artificially created missing group in ISIC, as otherwise DFR and GROUPDRO would not be applicable. We **bold** the highest WGA without any group annotation requirement and underline the best among all methods. We report the mean and standard deviation over 5 runs.

| Annot. Tr/Val | | CelebA | | ISIC | | Waterbirds | |
|---|---|---|---|---|---|---|---|
| | | AVG | WGA | AVG | WGA | AVG | WGA |
| ✗/✗ | ERM (baseline) | $90.1_{\pm0.3}$ | $49.7_{\pm1.0}$ | $86.9_{\pm0.2}$ | $34.4_{\pm0.8}$ | $91.5_{\pm0.9}$ | $74.9_{\pm2.8}$ |
| ✓/✓ | DFR (Kirichenko et al., 2023) | $91.3_{\pm0.3}$ | $88.3_{\pm1.1}$ | $87.4_{\pm1.3}^*$ | $\underline{77.7}_{\pm2.4}^*$ | $93.7_{\pm0.4}$ | $91.5_{\pm1.0}$ |
| ✓/✓ | GROUPDRO (Sagawa et al., 2020a) | $93.0_{\pm0.1}$ | $88.7_{\pm0.3}$ | $87.7_{\pm0.8}^*$ | $59.0_{\pm0.4}^*$ | $90.3_{\pm0.0}$ | $87.0_{\pm0.1}$ |
| ✗/✓ | CNC** (Zhang et al., 2022) | $93.9_{\pm0.1}$ | $88.9_{\pm1.3}$ | n.a. | n.a. | $92.0_{\pm0.6}$ | $89.9_{\pm0.6}$ |
| ✗/✓ | DEDIER** (Tiwari et al., 2024) | $93.2_{\pm0.1}$ | $89.6_{\pm1.7}$ | n.a | n.a | $92.1_{\pm0.4}$ | $89.8_{\pm0.5}$ |
| ✗/✓ | SELF**[6] (LaBonte et al., 2024) | n.a | $83.9_{\pm0.9}$ | n.a | n.a | n.a | $\underline{93.0}_{\pm0.3}$ |
| ✗/✓ | SPARE** (Yang et al., 2024) | $91.1_{\pm0.1}$ | $\underline{90.3}_{\pm0.3}$ | n.a | n.a | $96.2_{\pm0.6}$ | $91.6_{\pm0.8}$ |
| ✗/✗ | DCWP (Park et al., 2023) | $88.9_{\pm2.3}$ | $73.2_{\pm2.0}$ | $86.4_{\pm1.7}$ | $45.1_{\pm0.8}$ | $82.2_{\pm1.4}$ | $66.4_{\pm3.5}$ |
| ✗/✗ | DFR$_{\text{class}}$ | $84.9_{\pm2.1}$ | $67.7_{\pm0.8}$ | $74.1_{\pm2.6}$ | $48.8_{\pm6.0}$ | $91.0_{\pm0.8}$ | $75.5_{\pm1.2}$ |
| ✗/✗ | **PruSC** | $91.0_{\pm0.3}$ | $\mathbf{89.7}_{\pm0.3}$ | $86.1_{\pm0.4}$ | $\mathbf{75.1}_{\pm1.7}$ | $89.4_{\pm2.9}$ | $\mathbf{82.5}_{\pm2.2}$ |

**Hyper-parameter Setup.** We use ResNet18 and ResNet50 pre-trained on ImageNet. We train the models for 50 epochs using SGD with a constant learning rate of $10^{-3}$, momentum decay of 0.9, weight decay of $10^{-2}$, and batch size $32 - 64$. All images are resized to $224 \times 224$. We use ResNet50 to evaluate the predictive performance in Tab. 2 to compare with the original work for CNC, DEDIER and SELF. We use ResNet18 to investigate the robustness of our model to multiple spurious correlations (Sec. 6.5) and in the ablation study (Sec. 7.1). For the subnetwork extraction approaches (DCWP and PruSC), we report results with pruning around 50% of the network parameters and discuss the impact of other pruning ratios in Sec. 7.3.

## 6.3 Evaluation Metrics

We report **worst-group accuracy (WGA)**, the accuracy of the group (attribute-label combination) that a particular method performs worst. WGA is the standard metric for evaluating whether models use spurious correlations (Sagawa et al., 2020a). A significant drop between average accuracy and worst-group accuracy is a strong indicator of reliance on spurious correlations. To assess the impact on the group with the fewest training samples (usually the group with a spurious attribute uncorrelated with a target label), we report **minority-group accuracy (MGA)** for each spurious attribute $a_i$. The difference between WGA and MGA is that the former is defined per method, and the latter is defined on the data. Since the minority group is usually the group with a spurious attribute uncorrelated with a target label, performance is mostly lowest on this group and WGA=MGA, but not necessarily. To assess group fairness, i.e., whether all groups have similar prediction errors, we measure the **unbiased accuracy gap (UAG)**. Unbiased accuracy (**UA**) is the accuracy on a balanced test set, sampled such that there is no correlation between the attribute $a_i$ and a target label. UAG is the difference between the overall (official) test set accuracy and the UA. A small UAG indicates that groups are treated equally, regardless of whether they are minority or majority groups.

## 6.4 Predictive Performance

Our main results on predictive performance are shown in Tab. 2. PruSC shows the best performance in terms of worst-group accuracy (WGA) among methods that do not require any group annotations and shows comparable performance in most cases (CelebA and ISIC) to state-of-the-art methods that rely on annotations. Notably, our method successfully improves the accuracy of the worst group *malignant with patch* in ISIC, although it has never seen an example of this group during training. Our performance is even comparable to GROUPDRO and DFR which explicitly need (artificial) minority group samples during training.

Table 3: **Minority group** accuracy (MGA) across multiple spurious attributes $\mathcal{A} = \{$male, no beard, heavy makeup, wearing lipstick, young, pale skin$\}$ on CelebA with classification target hair color. ERM (baseline), DCWP and PRUSC are trained with class labels only, GROUPDRO and DFR in addition on attribute $a_i =$ male to mitigate the spurious correlation on gender. We report the accuracy of the minority groups defined in the training dataset. We **bold** the highest MGA among all methods.

| Annot. Tr/Val | | male | no beard | heavy makeup | wearing lipstick | young | pale skin |
|---|---|---|---|---|---|---|---|
| ✗/✗ | ERM (baseline) | $48.5_{\pm1.0}$ | $42.3_{\pm1.3}$ | $79.0_{\pm0.6}$ | $79.1_{\pm0.5}$ | $88.2_{\pm0.4}$ | $87.1_{\pm0.6}$ |
| ✓/✓ | GROUPDRO | $88.7_{\pm0.3}$ | $86.0_{\pm0.7}$ | $88.9_{\pm0.6}$ | $\mathbf{90.1}_{\pm0.5}$ | $89.8_{\pm0.8}$ | $88.2_{\pm0.3}$ |
| ✓/✓ | DFR | $86.1_{\pm0.2}$ | $78.0_{\pm0.5}$ | $83.9_{\pm0.4}$ | $83.9_{\pm0.4}$ | $86.0_{\pm0.1}$ | $81.8_{\pm0.2}$ |
| ✗/✗ | DCWP | $82.8_{\pm1.1}$ | $80.0_{\pm0.9}$ | $77.5_{\pm0.5}$ | $77.4_{\pm0.5}$ | $82.1_{\pm1.0}$ | $80.1_{\pm0.9}$ |
| ✗/✗ | **PruSC** | $\mathbf{91.0}_{\pm0.3}$ | $\mathbf{92.1}_{\pm0.2}$ | $\mathbf{89.9}_{\pm0.5}$ | $89.3_{\pm0.5}$ | $\mathbf{90.5}_{\pm0.1}$ | $\mathbf{89.3}_{\pm0.7}$ |

The $\text{DFR}_{\text{class}}$ method mimics DFR but replaces the group-balanced subset (which requires manual annotations) with a class-balanced subset (class annotations are readily available). $\text{DFR}_{\text{class}}$ falls short compared to DFR in all cases by a large margin, highlighting the importance of group annotations in DFR retraining. Using the same retraining dataset as $\text{DFR}_{\text{class}}$, the success of our method PRUSC emphasizes the necessity of distorting spurious clusters, which we achieve through a contrastive loss. We empirically analyze the impact of individual components of our method in an ablation study (Sec. 7.1).

On Waterbirds, there is a clear gap between annotation-free methods and those that rely on annotations. Using only annotations in the validation set, SELF even outperforms methods that require more annotations. The reason is a peculiarity of the Waterbirds dataset: its validation set is spurious-free, i.e., each class has equal quantities of land and water backgrounds, which gives an advantage to methods that rely on the validation set. This advantage becomes particularly apparent for $\text{DFR}_{\text{class}}$. By sampling a class-balanced subset from the validation set instead of the training set, LaBonte et al. (2024) report a WGA of 92.6 for $\text{DFR}_{\text{class}}$, which is comparable to the performance of DFR on a group-balanced subset obtained from label annotations. For truly annotation-free methods (such as ours) a spurious-free validation is not available, as its construction itself requires annotations.

## 6.5 Robustness to Multiple Spurious Correlations

In this section, we analyze how PRUSC performs in the presence of multiple spurious correlations within a dataset. Since we do not explicitly use information about the spurious attribute during subnetwork training, we expect the representations to be agnostic to any spurious feature. We use the CelebA dataset which has a potential set of spurious attributes $\mathcal{A}$ (Seo et al., 2022) (see Appendix A.3 for details). Specifically, we investigate six potential spurious attributes $a_i \in \mathcal{A} = \{$male, presence of beard, heavy makeup, wearing lipstick, young, pale skin$\}$. For each attribute $a_i$, we create a group set $\mathcal{G} = \{$(blond, $a_i$), (non-blond, $a_i$), (blond, non-$a_i$), (non-blond, non-$a_i$)$\}$. We train an ERM model to predict hair color and evaluate the performance separately for each group in $\mathcal{G}$. We then train all methods for correcting spurious attributes. GROUPDRO and DFR are trained on balanced groups for the attribute male.[7]

In Tab. 3, we report the performance of the minority group across multiple spurious attributes. Details of the data distribution and accuracy values for each individual group $\mathcal{G} = \{$(blond, $a_i$), (non-blond, $a_i$), (blond, non-$a_i$), (non-blond, non-$a_i$)$\}$ are provided in the Appendix, Tab. 7 and Tab. 9.

---

[6] LaBonte et al. (2024) report their method as annotation-free in their result tables, but state in the limitations section that they use "a small validation dataset with group annotations for model selection" and call for future work to completely remove this assumption.

[7] GROUPDRO and DFR are single-attribute methods, and cannot be trained for multiple attributes at the same time. Since the spurious attributes in CelebA are themselves strongly correlated (male, beard) there are still performance improvements for attribute groups not specifically trained for.

Table 4: Multiple spurious attributes. We report unbiased accuracy (UA), worst-group accuracy (WGA) and unbiased accuracy gap (UAG). We **bold** the highest WGA and underline the smallest UAG for each spurious attribute. ↑ higher is better, ↓ lower is better.

|  |  | male | no beard | heavy makeup | wearing lipstick | young | pale skin |
|---|---|---|---|---|---|---|---|
| WGA ↑ | ERM | $48.5_{\pm 1.0}$ | $42.3_{\pm 1.3}$ | $79.0_{\pm 0.6}$ | $79.1_{\pm 0.5}$ | $88.2_{\pm 0.4}$ | $87.1_{\pm 0.6}$ |
|  | GROUPDRO | $86.7_{\pm 0.7}$ | $86.0_{\pm 0.7}$ | $\mathbf{88.9}_{\pm 0.6}$ | $88.3_{\pm 0.5}$ | $\mathbf{85.7}_{\pm 0.4}$ | $87.4_{\pm 0.7}$ |
|  | PRUSC | $\mathbf{88.9}_{\pm 0.4}$ | $\mathbf{89.6}_{\pm 0.5}$ | $88.7_{\pm 0.1}$ | $\mathbf{88.7}_{\pm 0.2}$ | $82.0_{\pm 0.2}$ | $\mathbf{89.0}_{\pm 0.4}$ |
| UA ↑ | ERM | $82.0_{\pm 1.1}$ | $78.7_{\pm 0.5}$ | $88.4_{\pm 0.8}$ | $88.0_{\pm 0.8}$ | $91.3_{\pm 0.2}$ | $91.0_{\pm 0.3}$ |
|  | GROUPDRO | $88.5_{\pm 0.4}$ | $91.7_{\pm 0.2}$ | $91.0_{\pm 1.0}$ | $91.4_{\pm 0.2}$ | $90.5_{\pm 0.4}$ | $90.3_{\pm 0.3}$ |
|  | PRUSC | $90.4_{\pm 0.2}$ | $91.1_{\pm 0.2}$ | $90.0_{\pm 0.5}$ | $90.2_{\pm 0.3}$ | $90.0_{\pm 0.5}$ | $89.8_{\pm 0.1}$ |
| UAG ↓ | ERM | $7.0_{\pm 0.2}$ | $9.4_{\pm 1.1}$ | $0.9_{\pm 0.2}$ | $0.9_{\pm 0.2}$ | $2.0_{\pm 0.4}$ | $2.0_{\pm 0.3}$ |
|  | GROUPDRO | $2.2_{\pm 0.2}$ | $2.0_{\pm 0.2}$ | $0.8_{\pm 0.1}$ | $1.4_{\pm 0.2}$ | $1.0_{\pm 0.3}$ | $1.1_{\pm 0.1}$ |
|  | PRUSC | $\underline{0.5}_{\pm 0.1}$ | $\underline{1.0}_{\pm 0.1}$ | $\underline{0.1}_{\pm 0.0}$ | $\underline{0.1}_{\pm 0.0}$ | $\underline{0.8}_{\pm 0.1}$ | $\underline{0.3}_{\pm 0.0}$ |

The ERM models usually fail for the minority group represented by the target and uncorrelated attributes. For example, for the attribute gender = male, the ERM model only has 48.5% accuracy for the group (blond hair, male) since in the training set there are only $1,102$ samples in the group (blond hair, male) while the group (non-blond hair, male) contains $53,483$ samples. For the ERM model in Tab. 3, the minority group accuracy is indeed always equal to worst group accuracy, which is not the case for the other methods. For example, GROUPDRO has an accuracy of 88.7% on the minority group (blond hair, male), but worst-group accuracy of 86.7% for the group (not blond hair, not male). PRUSC has the largest improvement over the baseline across all spurious attributes. In particular, on the spurious attribute "male", PRUSC even outperforms GROUPDRO and DFR which were trained with known group labels for this attribute.

In Tab.4 we compare GROUPDRO that relies on group annotations, with PRUSC that does not require prior knowledge of spurious attributes. Both methods show an increase of worst-group accuracy (WGA) over the ERM baseline for all groups, suggesting that both methods can successfully mitigate certain spurious correlations. However, in CelebA's official training and test split, the groups are identically distributed. The WGA indicates whether a method has successfully reduced the impact of biasing features, but because the average accuracy is driven by the majority group, it may mask performance declines in the remaining groups. That is, we cannot determine from these two measures alone whether a model treats all groups equally. Our approach, PRUSC, is on par with GROUPDRO in terms of unbiased accuracy (UA) and WGA. This indicates that both methods can effectively unlearn spurious correlations and maintain strong performance even when there is a distribution shift from a highly imbalanced training set to a balanced test set. However, PRUSC shows a smaller UGA in all cases and is therefore more consistent across different spurious attributes and shows a more balanced performance across groups.

# 7 Detailed Analysis

In this section, we show an ablation study to evaluate the effectiveness of PRUSC components, including the pruning and fine-tuning steps, as well as the use of contrastive loss (Sec.7.1). We also analyze the impact of number of clusters (Sec.7.2), the impact of varying pruning ratios (Sec.7.3), the performance with different choices of contrastive batch in PRUSC (Sec.7.4) and the representation space after mitigation (Sec.7.5).

## 7.1 Component Ablation

We analyze the impact of each component of PRUSC. Tab. 5 shows an overview of our settings. Specifically, we evaluate the impact of pruning, i.e., subnetwork extraction and fine-tuning on performance, along with the effect of applying contrastive loss at each step. If the contrastive loss is omitted during pruning (ConLoss Pr ✗), we train the subnetwork with cross-entropy loss and sparse regularization. Without the contrastive loss

Table 5: Ablation study on CelebA (target: blond hair, spurious attribute: male). Effect of pruning (Pr) and finetuning (Ft) steps with (✓) and without (✗) our contrastive loss, cells marked 'n.a.' show no loss values because there is no corresponding pruning or fine-tuning stage. $\checkmark_{last}$ means fine-tuning last layer only.

| Setting | Fine-tuning | ConLoss Pr | ConLoss Ft | WGA | AVG | UA |
|---|---|---|---|---|---|---|
| | WITH SUBNETWORK EXTRACTION | | | | | |
| 1 | ✓ | ✓ | ✓ | 67.7 | 83.7 | 80.4 |
| 2 (PRuSC) | ✓ | ✓ | ✗ | **89.6** | **90.1** | **90.5** |
| 3 | ✓ | ✗ | ✗ | 41.8 | 91.5 | 73.9 |
| 4 | ✗ | ✓ | n.a. | 43.7 | 89.6 | 74.4 |
| | WITHOUT SUBNETWORK EXTRACTION | | | | | |
| 5 | ✓ | n.a. | ✓ | 67.4 | 88.8 | 81.6 |
| 6 | ✓ | n.a. | ✗ | 78.9 | 87.9 | 87.2 |
| 7 (DFR$_{class}$) | $\checkmark_{last}$ | n.a. | ✗ | 79.5 | 83.8 | 85.5 |

during fine-tuning (ConLoss Ft ✗), we fine-tune the subnetwork or full model with cross-entropy loss alone. In setting 1 - 4, 'Fine-tuning' column refers to fine-tuning for 2 to 5 epochs after extracting subnetwork.

**Subnetwork Extraction.** Settings 5 to 7 investigate whether one can improve the worst-group accuracy only by fine-tuning and take advantage of designing a class-balanced subset ($\mathcal{D}_{task}$) without a subnetwork. We skip the *Binary Mask Training* step (cf. Sec. 5.3) and either fine-tune the whole network (settings 5 and 6), or only the last layer (setting 7) for the same number of epochs with $\mathcal{D}_{task}$. The latter setting is equivalent to DFR$_{class}$, with $\mathcal{D}_{task}$. We observe no significant improvement when fine-tuning the whole network compared to fine-tuning only the last layer (setting 6 vs. 7), fine-tuning the whole network even results in worse WGA.

**Fine-tuning.** Without fine-tuning after subnetwork extraction (setting 4) the performance of the subnetwork drops significantly, to similar performance as before pruning. This finding aligns with previous work on modular subnetworks (Csordás et al., 2020; Zhang et al., 2021; Park et al., 2023), which require retraining or fine-tuning the extracted network after pruning to achieve better performance.

**Contrastive Loss during Pruning.** Settings 2 and 3 show the effect of the contrastive loss during pruning (ConLoss Pr), which is the key feature of PRuSC. In setting 3, we prune the networks using the cross-entropy loss and sparse regularization, i.e., vanilla pruning. Vanilla pruning leads to slightly better AVG but keeps the WGA the same as before pruning, while PRuSC (setting 2) significantly increases the WGA, showing that the contrastive loss helps to find the *correct* subnetwork that is robust to spurious correlations. The contrastive loss forces the pruning process to retain only those connections that contribute to an optimized representation, ensuring that all spurious attributes are distributed in the representation space.

**Contrastive Loss during Fine-tuning.** Fine-tuning with contrastive loss (ConLoss Ft) reduces the WGA significantly compared to fine-tuning without contrastive loss, both with (setting 1 vs. 2) and without subnetwork extraction (5 vs. 6 and 7). In all those settings, we fine-tune with the balanced subset $\mathcal{D}_{task}$. Kirichenko et al. (2023) showed that ERM models can effectively learn all features, but spurious features contribute more strongly to the prediction, due to the data distribution. Hence, during fine-tuning without subnetwork extraction, the model only has to adjust the classifier weights to rely more on invariant features. This is also confirmed by the increased WGA in setting 7 (fine-tuning only the last layer vs. fine-tuning all layers in setting 6). Adding the contrastive loss (setting 5), the model now has to optimize a dual objective: classification and representation, which is likely confusing rather than helpful, in particular as the balanced subset is comparably small. Similarly, adding the contrastive loss during fine-tuning after the extraction of a sub-network (setting 1) hurts performance. In this setting, the representation space has been optimized by the extraction of the subnetwork already and fine-tuning is only supposed to adjust the weight magnitudes after pruning. We hypothesize that forcing the network to also re-learn the representation space does not effectively align the remaining weights, but rather distorts weights and representations.

**Contrastive Learning with (pseudo) Group Labels.** Instead of using contrastive learning with clusters defined by representation clustering, we can use annotated group labels. The clusters in $\mathcal{D}_{\text{task}}$ by ground truth group labels correspond to spurious attributes, i.e. the two clusters in CelebA with respect to the spurious attribute gender. This approach does significantly improve the WGA (from 50.2% to 84.8%, see Tab. 10 in the Appendix), highlighting the potential of using annotated or pseudo-group labels. However, contrastive learning with ground truth labels does not outperform PruSC with representation clustering (WGA 89.6%). We hypothesize that this is due to the alignment between our contrastive loss and contrastive batch sampling, which is particularly beneficial in the representation space. Further results and discussion are presented in App. A.5.

## 7.2 Impact of the Number of Clusters

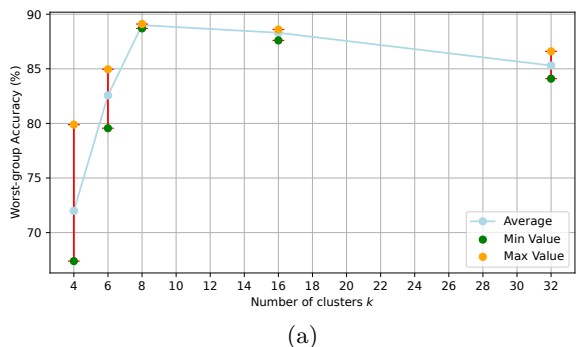
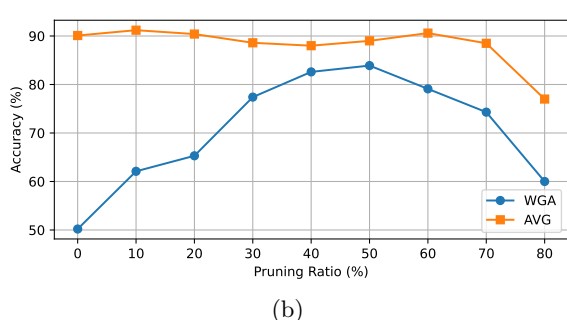

(a)

(b)

Figure 4: Analysis of PruSC wrt. the number of clusters and pruning ratio on the CelebA dataset. (a) Average, minimum and maximum worst group accuracy (WGA) among 5 runs across varying numbers of clusters. (b) WGA and average accuracy (AVG) of varying pruning ratios.

We analyzed the impact of the number of clusters $k$ in $k-$means clustering on the CelebA dataset. Fig. 4a shows the average WGA together with minimum and maximum among 5 runs across different choices of $k$. For $k < 8$ performance is unstable and generally low. The effectiveness of PruSC depends on the purity of clusters with respect to classes and spurious features. For lower values of $k$, both, class purity ($\mathcal{Y}$-purity) and spurious purity ($\mathcal{A}$-purity) are also lower (cf. $k = 4$ vs $k = 8$, Tab. 1). With lower cluster purity, it is more likely that each cluster contains samples from all groups in $\mathcal{G}$, making it harder for our method to sample a more group-balanced subset, which in turn affects WGA performance. For $k \geq 8$, the performance of PruSC remains rather stable, indicating that $k$ is not a critical hyper-parameter of our method, as long as it is sufficiently large to obtain pure clusters, both in terms of classes and spurious features.

## 7.3 Impact of the Pruning Ratio

Fig. 4b shows the average and worst-group accuracy of our model on CelebA for different pruning ratios. We test on the standard setup of classifying hair color and considering gender as spurious attribute. While the average accuracy does not change much as long as the pruning ratio is below 70%, the worst-group accuracy varies more significantly. Subnetworks in the regime of a small gap between average and worst-group accuracy are less prone to spurious correlations. On the CelebA dataset, the most effective pruning ratio is around 40% to 50%. When pruning beyond 70% of the parameters, the model seems to fail to predict accurately, as both, worst-group and average accuracy drop.

## 7.4 Choice of Positive and Negative Samples in Contrastive Learning

We study the effectiveness of our sampling choices for contrastive batch learning in PruSC. Recall our default setting: for a random *anchor*, we sample *positives* from different clusters having the same class label as the *anchor* and *negatives* from the same cluster (regardless of class label). We compare this choice with sampling *negatives* from the same cluster, but different class labels (Negative Ablation) and classic

supervised contrastive learning (SupCon) (Khosla et al., 2020), where *positives* and *negatives* are defined only by class labels. The results in Tab. 6 show that both choices are less effective w.r.t. worst-group accuracy.

Sampling *negatives* from the same cluster as the *anchor* but with a different class label (Negative Ablation) is affected by the purity of clusters. As shown in Fig. 2 and Tab. 1, $k-$means clustering on the representations of a converged ERM model leads to high purity clusters in terms of class labels. Therefore, a cluster may not contain a sufficient number of *negatives* with different class labels. Early stopping could solve this problem, but requires access to the model training procedure, and requires additional optimization of the stopping criteria to balance model performance and the amount of misclassifications.

Table 6: Average accuracy (AVG) and worst-group accuracy (WGA) of PRUSC with different sampling choices for the contrastive batch.

|  | AVG | WGA |
|---|---|---|
| Default | 91.0% | 89.7% |
| Negative Ablation | 90.2% | 73.1% |
| SupCon (Khosla et al., 2020) | 92.2% | 61.5% |

Instead, SupCon aims to bring the representations of the same class closer together, tightening the samples within the same cluster, which has a high potential to increase the spurious correlation instead of reducing it. Furthermore, SupCon primarily helps to separate classes (cf. increased average accuracy), which ERM models may already do sufficiently well (cf. Fig. 2, left and Fig. 5, left). Our contrastive batch is defined not only by class labels as in SupCon, but also by the positions of the samples in the representation space based on clustering.

### 7.5 Representation Space after Mitigating Spurious Correlations

In this section, we discuss how the mitigation strategy of DCWP and PRUSC affects the representation space. In Sec. 5.1 we showed that ERM tends to learn and cluster samples based on spurious attributes during training (Fig. 2). In Fig. 5 (left), we visualize the embedding space of the non-blond hair class in the test set of the same ERM model, which reveals again ERM learns based on spurious attributes. Despite achieving high average test accuracy (90%), the fact that ERM relies on spurious correlations in its predictions leads to a low worst-group accuracy (49.7%) and lack of generalization ability.

DCWP mitigates spurious correlations by extracting a subnetwork by training a contrastive loss, and sampling positives and negatives from bias-conflict (early-stop ERM misclassified) and bias-align (early-stop ERM correctly classified) cases. In terms of performance measures, DCWP successfully improves worst-group accuracy (73%), but the model still tends to group samples by the female attribute (Fig. 5, center). We hypothesize that defining bias-conflict based on ERM misclassified cases is not reliable. When a spurious attribute (female) is easy to learn and strongly correlated with a class (blond hair), the model learns easily and predicts correctly based on the spurious attribute, even in the early stages of training.

PRUSC eliminates this reliance on spurious feature learning. While the average accuracy remains high (91%) through cross-entropy loss, the use of contrastive learning with a distortion of spurious clusters in the representation space eliminates the reliance on spurious features and thus improves generalization. Our pruned model mixes up samples with different attributes within one class showing its independence from these spurious attributes (Fig. 5, right). Additional visualizations are shown in Appendix A.4.

## 8 Conclusion

In this paper, we introduced PRUSC (Pruning Spurious Correlations) to extract a spurious-free subnetwork from a fully trained dense neural network. Our method does not rely on annotations of spurious attributes, but instead optimizes the representation space to produce class-induced clusters rather than clusters induced by spurious attributes (as observed in the representation space of the dense network trained with ERM).

PRUSC achieves competitive worst-group accuracy across several benchmarks *without requiring any annotation of spurious features at all* (not even in the validation set). While PRUSC outperforms all annotation-free methods in comparison, we found that the spurious-free validation set of the Waterbirds dataset might lead to an overestimation of the performance of methods that rely on annotations in the validation set. Our

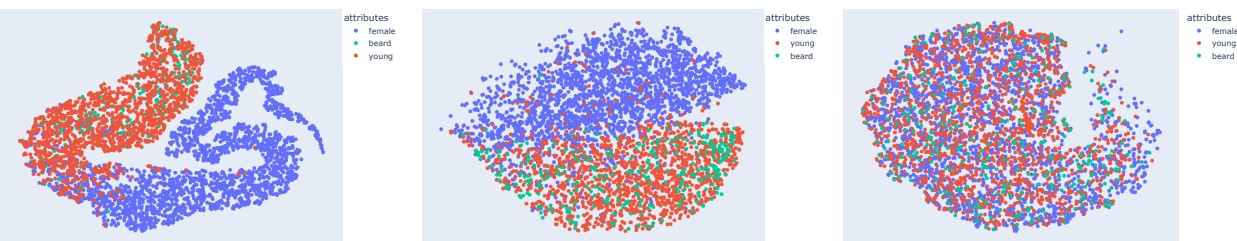

Figure 5: Test set representation space on CelebA for predicting hair color. The visualization shows data points of the class 'non-blond hair', colors represent the spurious attributes. All models achieve high average test accuracy (over 88%, see Tab. 2). While ERM (left) and DCWP (center) cluster samples by spurious attributes, PRuSC (right) mixes samples with these attributes, obtaining a better worst-group accuracy.

truly annotation-free approach not only eliminates the annotation overhead and makes PRuSC applicable to implicit and unknown spurious features, but also results in subnetworks that are robust to multiple spurious correlations.

By extracting the subnetwork from a fully trained network without extensive retraining of the weights, we show that even dense networks, that are heavily influenced by spurious correlations, contain smaller parts that learn invariant attributes relevant to the task. In future work, we plan to analyze the extracted subnetworks in detail, in order to quantify the information (encoded features and relations) that is retained and also the information that is lost. In particular, we aim to identify whether relevant information is lost and how to retain it (and vice versa for irrelevant information) in order to further improve performance.

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

# A  Appendix / supplemental material

## A.1  Datasets

**CelebA.**  Tab. 7 shows the distribution of group $\mathcal{G} = \{(\text{blond}, a_i), (\text{non-blond}, a_i), (\text{blond}, \text{non-}a_i), (\text{non-blond}, \text{non-}a_i)\}$ in training dataset and testing dataset. This shows the imbalanced distribution of the group (target, correlated attribute) and (target, un-correlated attribute). The models can both depend on the easier to learn features and the dominance of training data to suffer from spurious correlations. Generally, the minority group (group having smallest portion in training set) is the group having worst accuracy in test (with respect to baseline ERM).

Table 7: Data distribution in CelebA dataset of groups $g \in \mathcal{G} = \mathcal{Y} \times \mathcal{A}$ forming by target class blond or non-blond hair $y \in \mathcal{Y}$ and spurious attribute $a \in \mathcal{A}$. Gray color highlights the **minority group** - smallest $p(g)$ among groups in the training set.

| Attribute $a$ | Group $g$ | Training distribution | | | Testing distribution | | |
|---|---|---|---|---|---|---|---|
| | | Quantity | $p(g)$ (%) | $p(t \mid a)$ (%) | Quantity | $p(g)$ (%) | $p(t \mid a)$ (%) |
| male (M) | $y=0, a=0$ | $1,558$ | 2.1 | 7.8 | $14,415$ | 44.3 | 76.4 |
| | $y=0, a=1$ | $53,483$ | 71.7 | 98.0 | $13,391$ | 41.1 | 97.9 |
| | $y=1, a=0$ | $18,417$ | 24.7 | 92.2 | $4,463$ | 13.7 | 23.6 |
| | $y=1, a=1$ | $1,102$ | 1.5 | 2.0 | $285$ | 0.9 | 2.1 |
| no beard (NB) | $y=0, a=0$ | $21,229$ | 28.5 | 98.9 | $5,369$ | 16.5 | 98.8 |
| | $y=0, a=1$ | $33,812$ | 45.3 | 63.7 | $22,437$ | 68.9 | 82.7 |
| | $y=1, a=0$ | $243$ | 0.3 | 1.1 | $64$ | 0.2 | 1.2 |
| | $y=1, a=1$ | $19,276$ | 25.9 | 36.3 | $4,684$ | 14.4 | 17.3 |
| heavy makeup (HM) | $y=0, a=0$ | $53,885$ | 72.3 | 89.5 | $18,620$ | 57.2 | 92.3 |
| | $y=0, a=1$ | $1,156$ | 1.6 | 8.1 | $9,186$ | 28.2 | 74.2 |
| | $y=1, a=0$ | $6,317$ | 8.5 | 10.5 | $1,550$ | 4.8 | 7.7 |
| | $y=1, a=1$ | $13,202$ | 17.7 | 91.9 | $3,198$ | 9.8 | 25.8 |
| wearing lipstick (WL) | $y=0, a=0$ | $53,480$ | 71.7 | 93.6 | $16,386$ | 50.3 | 94.9 |
| | $y=0, a=1$ | $1,561$ | 2.1 | 9.0 | $11,420$ | 35.1 | 74.7 |
| | $y=1, a=0$ | $3,678$ | 4.9 | 6.4 | $880$ | 2.7 | 5.1 |
| | $y=1, a=1$ | $15,841$ | 21.2 | 91.0 | $3,868$ | 11.9 | 25.3 |
| young (Y) | $y=0, a=0$ | $16,262$ | 21.8 | 31.4 | $6,473$ | 19.9 | 89.2 |
| | $y=0, a=1$ | $3,257$ | 4.4 | 14.3 | $21,333$ | 65.5 | 84.3 |
| | $y=1, a=0$ | $35,456$ | 47.6 | 68.6 | $780$ | 2.4 | 10.8 |
| | $y=1, a=1$ | $19,585$ | 26.3 | 85.7 | $3,968$ | 12.2 | 15.7 |
| pale skin (PS) | $y=0, a=0$ | $53,664$ | 72.0 | 74.7 | $26,758$ | 82.2 | 85.8 |
| | $y=0, a=1$ | $1,377$ | 1.8 | 49.9 | $1,048$ | 3.2 | 76.1 |
| | $y=1, a=0$ | $18,139$ | 24.3 | 25.3 | $4,418$ | 13.6 | 14.2 |
| | $y=1, a=1$ | $1,380$ | 1.9 | 50.1 | $330$ | 1.0 | 23.9 |

**Waterbirds.**  Waterbirds is an artificial dataset generated based on the code from (Sagawa et al., 2020a). Similar to the CelebA dataset, the training set of Waterbirds suffers from both high class imbalance and significant spurious correlations, with waterbirds typically appearing in water backgrounds and landbirds in land backgrounds. Interestingly, the validation and test sets of Waterbirds are balanced in terms of these spurious correlations, with equal numbers of waterbirds and landbirds appearing in both land and water backgrounds. Therefore, we expect that all methods tuning hyper-parameters on the validation set will have access to distribution information reflective of the test set, leading to high overall performance and improved worst-group accuracy. We show details number of samples in each group in Tab. 8.

Table 8: Distribution of samples in dataset Waterbirds and ISIC. Gray color highlights the minority group. Waterbirds has similar distribution between validation and test sets, and they are both spurious-free within a class (i.e., equal number of samples with land or water background). In ISIC, we construct a group balanced validation set for the purpose of training DFR which requires a held-out group-balanced dataset.

| Dataset | Groups | Train | Validation | Test |
|---|---|---|---|---|
| Waterbirds | waterbird on water | 1057 | 133 | 642 |
| | waterbird on land | 56 | 133 | 642 |
| | landbird on water | 184 | 466 | 2255 |
| | landbird on land | 3498 | 467 | 2255 |
| ISIC | Benign with patch | 5526 | 60 | 2763 |
| | Benign without patch | 6314 | 60 | 3158 |
| | Malignant with patch | 0 | 60 | 821 |
| | Malignant without patch | 1571 | 60 | 821 |

## A.2 Task-oriented Subnetworks

Csordás et al. (2020) aims to highlight sets of non-shared weights solely responsible for individual class $k$ by training a mask with class $k$ removed from the training dataset ($\mathcal{D}_{\text{task}} = \mathcal{D} \setminus \{(x, y | y = k)\}$) and need not add any other loss term. The weights solely responsible for this class will be absent from the resulting mask. Results show that the performance of the target class drops significantly, while only a small drop in performance is observed for non-target classes. This indicates a heavy reliance on class-exclusive, non-shared weights in the feature detectors. Interestingly, Csordás et al. (2020) also shows the misclassification behaviors based on "shared features" (or possibly spurious features), e.g., when "airplane" is removed, images are often misclassified as "birds" or "ships," both of which commonly feature a blue background.

Zhang et al. (2021) investigate the capability of finding a subnetwork for a specific task within a trained dense network by extracting it using a specially designed held-out dataset tailored for that task. Specifically, the authors train a dense network on the Colored MNIST dataset, which contains a strong spurious correlation between digits and colors, with each digit being strongly correlated with a particular color. Using the same technique as Csordás et al. (2020), but while pruning, Zhang et al. (2021) train the network with a dataset balanced in terms of colors and digits to isolate subnetworks that solely predict either digits or colors. Surprisingly, even though the trained dense network suffers significantly from spurious correlations, it is still possible to extract a subnetwork that predicts only the digits with high accuracy without extensive retraining on a balanced (or non-spurious) dataset.

Park et al. (2023) construct the mask training data by upweighting the complex cases (called by bias-conflict samples) defined by the misclassified cases of the ERM model, aiming to extract a set of weights more robust to those cases, therefore eliminating bias in the network.

## A.3 Multiple spurious attributes

**CelebA – Dataset contains multiple spurious attributes.** We analyze 7 chosen attributes having annotations from dataset, namely: male, no beard, heavy makeup, wearing lipstick, young, pale skin, big nose. Treating each attribute independently, we show their potential of being spurious attribute by comparing the worst-group accuracy from set set $\mathcal{G} = \{(\text{blond}, a_i), (\text{non-blond}, a_i), (\text{blond}, \text{non-}a_i), (\text{non-blond}, \text{non-}a_i)\}$ for each attribute $a_i$ and the ERM average accuracy. A fair model should treat all groups equally and minimize the influence of distribution. As shown in Tab. 9 (cf. ERM), we see some gaps between worst-group and average accuracy, and at the same time, the unequal accuracy among groups.

**Model performance across multiple spurious attributes.** Tab. 9 shows more details on the accuracies of each group and compares among more methods, namely ERM, DCWP, GROUPDRO, DFR, and PRUSC. Interestingly, when considering the group having the worst performance by ERM, our method outperforms

all others by considerable margins. We argue that our cluster sampling stage can find hard cases even more effectively than methods using ERM failures such as DCWP.

Table 9: Details of group accuracy (%) with respect to multiple attributes in CelebA dataset. Average accuracy of baseline ResNet18 is 89.4%. After pruning, our subnetwork has keep-ratio 45.5% of ResNet18 weights. We color the column corresponding to the minority group with respect to the combination of target and spurious attribute.

| | T = blond hair | {T=0, A=0} | {T=0, A=1} | {T=1, A=0} | {T=1, A=1} |
|---|---|---|---|---|---|
| male | ERM | 78.5 | 98.9 | 98.9 | 50.2 |
| | DCWP | 75.9 | 92.7 | 97.0 | 81.8 |
| | **PruSC** | 90.5 | 90.8 | 89.6 | **91.2** |
| | GROUPDRO | 88.5 | 91.1 | 93.5 | 88.8 |
| | DFR | 84.9 | 85.6 | 94.9 | 85.6 |
| no beard | ERM | 99.4 | 85.7 | 37.5 | 96.7 |
| | DCWP | 93.9 | 81.6 | 78.1 | 96.4 |
| | **PruSC** | 92.3 | 89.6 | **92.2** | 89.7 |
| | GROUPDRO | 95.8 | 89.1 | 85.9 | 94.9 |
| | DFR | 90.4 | 82.4 | 78.1 | 96.6 |
| heavy makeup | ERM | 92.6 | 79.6 | 88.9 | 99.3 |
| | DCWP | 87.4 | 76.9 | 92.7 | 97.8 |
| | **PruSC** | 88.9 | **92.7** | 89.2 | 89.9 |
| | GROUPDRO | 89.4 | 90.6 | 91.5 | 94.1 |
| | DFR | 84.4 | 83.1 | 93.8 | 97.6 |
| wearing lipstick | ERM | 94.6 | 79.4 | 82.1 | 99.1 |
| | DCWP | 88.9 | 76.9 | 90.0 | 97.5 |
| | **PruSC** | 88.7 | **92.2** | 89.8 | 89.7 |
| | GROUPDRO | 89.6 | 89.9 | 90.0 | 94.0 |
| | DFR | 84.9 | 82.6 | 91.8 | 97.4 |
| young | ERM | 90.9 | 87.5 | 92.1 | 96.7 |
| | DCWP | 84.7 | 83.7 | 95.3 | 96.3 |
| | **PruSC** | 81.4 | **92.8** | 94.1 | 88.8 |
| | GROUPDRO | 84.5 | 91.3 | 95.1 | 92.9 |
| | DFR | 79.1 | 85.4 | 96.0 | 96.4 |
| pale skin | ERM | 88.5 | 83.5 | 95.8 | 97.9 |
| | DCWP | 84.1 | 79.2 | 96.2 | 95.2 |
| | **PruSC** | 90.2 | **88.7** | 89.6 | 90.6 |
| | GROUPDRO | 89.8 | 87.5 | 93.3 | 92.7 |
| | DFR | 84.0 | 81.9 | 96.4 | 96.1 |
| big nose | ERM | 85.8 | 95.3 | 96.7 | 86.9 |
| | DCWP | 81.3 | 91.4 | 96.2 | 94.4 |
| | **PruSC** | **90.4** | 89.4 | 89.4 | 93.3 |
| | GROUPDRO | 89.2 | 91.3 | 93.1 | 94.7 |
| | DFR | 82.8 | 87.2 | 96.3 | 96.9 |

## A.4 Feature Visualization

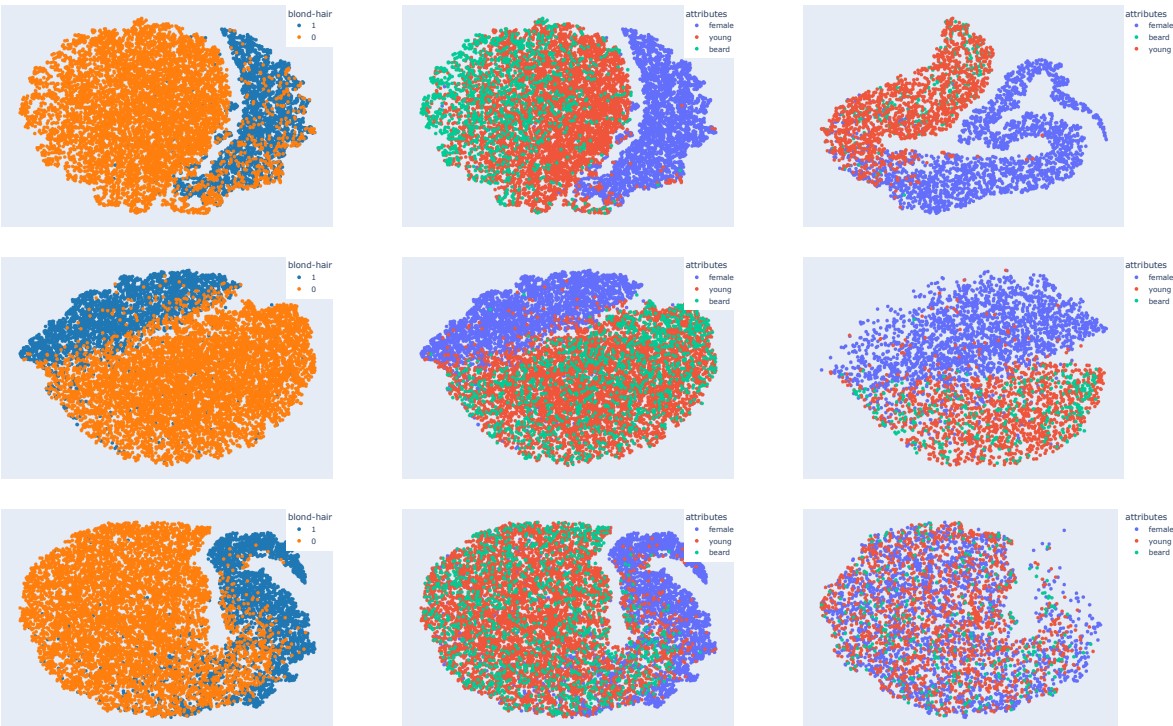

Figure 6: Embedding space of ERM (first row), DCWP (second row), and PRUSC (third row) on CelebA for predicting blond hair color. We visualize the train set embedding space with colors representing the class of non-blond hair (left column) and three other attributes: female, young, and beard (center column). To show how the approach works on test set, we visualize the class non-blond hair of test set, coloring above spurious attributes (right column). In both train and test embedding, ERM clusters well spurious attributes even those attribute labels are not used during training; showing its priority to learn spurious features. DCWP leveraging the ERM-misclassified cases can not mitigate the influence of attribute *female*, shown by a high purity cluster of samples with attribute female in both train and test representation space. Our pruned model mixes up samples in the same class, eliminating the biased learning behavior.

### A.5 PruSC with (pseudo) Group Labels

In this section, we study the effectiveness of PRUSC with annotated spurious attributes and other pseudo group labeling approaches such as using *minority and majority* groups based on ERM predictions Nam et al. (2020); Park et al. (2023); Zhang et al. (2022).

We conduct experiments on the CelebA dataset using the same frozen ERM model $f$ trained as defined in Sec. 5.3. The hyper-parameters remain identical to those used for training the original PRUSC. Regarding the changes, first, we do not perform representation clustering and sub-sampling. Instead, we design $\mathcal{D}_{\text{task}}$ to be (pseudo) group-balanced and fine-tune the model after pruning with this $\mathcal{D}_{\text{task}}$. Second, the contrastive loss defined during the binary mask training stage remains the same, but with the new *clusters* definition, as described in the following:

- **Annotated group labels.** Each cluster is defined based on the spurious attribute (gender), resulting in exactly two clusters for CelebA. We sample an equal number of data points from each group $\mathcal{G} = \mathcal{Y} \times \mathcal{A}$ in the training set, creating a group-balanced $\mathcal{D}_{\text{task}}$.

- **Pseudo labels by ERM predictions.** Following previous work (Nam et al., 2020; Park et al., 2023; Zhang et al., 2022), we track the ERM predictions on the training set using an early epoch checkpoint during the training process (early-stopped ERM). Samples misclassified by the early-stopped ERM form the *minority* group, which is assumed to be less influenced by spurious correlations, while the remaining samples form the *majority* group. We construct $\mathcal{D}_{\text{task}}$ by sampling an equal number of data points from each combination of groups, {minority, majority} $\times \mathcal{Y}$. There are exactly two clusters defined by the *minority* and *majority* groups, and $\mathcal{D}_{\text{task}}$ is also constructed class balanced.

Table 10: Details of group accuracy (%) in CelebA dataset when applying PRUSC with different defined clustering methods for constructing $\mathcal{D}_{\text{task}}$. Average accuracy of baseline ResNet18 is 89.4%. After pruning, subnetworks has keep-ratio around 50% of ResNet18 weights.

|  | $\mathcal{G}_1$ | $\mathcal{G}_2$ | $\mathcal{G}_3$ | $\mathcal{G}_4$ |
|---|---|---|---|---|
| *Baselines* | | | | |
| ERM | 78.5 | 98.9 | 98.9 | 50.2 |
| GROUPDRO (fine-tuning on balanced subset) | 88.5 | 91.1 | 93.5 | 88.8 |
| *PRUSC's contrastive loss with pseudo-labels* | | | | |
| Annotated group labels | 84.8 | 90.4 | 94.8 | 87.7 |
| ERM predictions | 68.7 | 92.3 | 91.4 | 64.9 |
| PRUSC | 90.5 | 90.8 | 89.6 | 91.2 |

Table 10 shows the behavior of our subnetwork extraction method with different clustering definitions. The results show that our method can be used in combination with a wide range of clustering methods, all of which improve the worst group accuracy. In particular, the contrastive loss is specifically designed to *push apart* samples with identical spurious attributes, i.e., samples grouped within the same cluster. Consequently, clusters defined by ERM predictions show limited improvement, as ERM misclassifications tend to divide the dataset into broad *minority* and *majority* groups, failing to effectively distinguish between different spurious attributes. On the other hand, clusters defined by ground truth group labels significantly increase the worst-group accuracy (50.2% to 87.7%). Furthermore, the subnetwork achieves similar group performance when applying GROUPDRO from scratch.

We hypothesize that the success of PRUSC comes from the alignment between the clustering method and the defined contrastive loss. The contrastive loss aims at reshaping the latent space by pulling and pushing samples according to specific criteria. In this context, defining *positive* and *negative* samples by representation distances poses the best choice as it directly reinforces the representation structure learned through the contrastive objective. Meanwhile, other sampling approaches that do not use the representation distance information could disrupt this alignment.

