# OpenReview forum: "Out of Spuriousity: Improving Robustness to Spurious Correlations without Group Annotations"
_TMLR — Accepted by TMLR_

### Review · Reviewer_54Z3 · 2024-10-15

**Summary Of Contributions:**

## Main contribution
The authors propose a new method for the training of neural networks such that
the network does not rely on spurious correlations. Networks that do rely on
these spurious correlations often have poor generalizability. Reducing the
reliance on spurious correlations would thus result in better performance.

Current methods that try to solve the same problem rely on the presence of
group labels. However acquiring these labels can be costly or requires
knowledge of what spurious correlations could exist. The method that the author
propose circumvents this reliance on group labels and requires no ground-truth
group annotations. The authors clearly motivate their approach and shed light
on their thought process when developing this method. They also supply
extensive background information of previous proposals and the different methods
they have employed when developing their method.

The experiments and the code provided
show the performance of their method. The results show that their method
beats several other baseline
methods on standard datasets in the field. This method looks like a valuable
contribution to battling the problem of spurious correlations in
classification. Finally, the authors provide an ablation analysis showing
that each of the components in their method actually contributes to the performance.

## The method
The authors summarise their method in 3 steps. Given a trained network $f$ the
following steps are performed:

1. The representations of the penultimate layer are calculated and clustered.
These clusters are labelled as $i$-dominant or neutral, depending on if class
$i$ is presented more than $90\%$ of the time in that cluster. From these
representations and cluster labels, a class-balanced dataset is sampled which
should result in a dataset that is $10\%$ the number of samples of the original
dataset

2. The weights of the network are frozen and a binary mask over the original
weights is learned. This binary mask is learned through a contrastive learning
approach, while trying to maintain a good performance on the original task

3. After the mask is learned, the weights of the subnetwork are again finetuned
using the original dataset and a cross-entropy loss.

## Experiments

The authors provided experiments on 3 datasets: The CelebA dataset, the
Waterbirds dataset and the ISIC Skin dataset. Their method is compared with 2
types of baseline models: 2 methods that require group annotations and 2
methods that do not need group annotations.

Various different metrics are reported to measure the performance with respect
to different goals. The reliance on spurious correlations is measured by the
worst-group accuracy (WGA), the performance on the smallest group is measured
by minority-group accuracy (MGA) and the group fairness is measured by the
Unbiased Acurracy (UAG).

Looking at the WGA results. It can be see that the method of the author
performs best amongst all the methods that do not need any group annotations.
The methods that do need group annotations perform better in some cases only.

The method by the author also seems to perform the best amongst their baselines
with respect to the MGA metric

Finally, the authors show that their method also performs well using the UAG
metric.

## Ablation

In the ablation study the authors systematically analyse all the components
of the their method. Specifically, they show that the pruning, contrastive learning
and finetuning after pruning all improve the WGA AVG and UA metrics.

Furthermore, they investigate the influence of their samplings scheme for the
positive and negative samples in the contrastive learning step.

Finally, it is shown empirically that the spurious features are mixed using
t-SNE visualization of the representations in the penultimate layer.

**Audience:**

Yes

**Broader Impact Concerns:**

I believe no broader Impact concern is needed.

**Claims And Evidence:**

Yes

**Requested Changes:**

# Requested Changes

- In the experiment section, the WGA scores are reported over multiple runs using the mean
  and standard deviation, while the
  MGA and UAG scores are only reported using single runs. It would be good to see
  the results of these metrics over multiple runs as well and with the mean and standard deviation.

- There is a related work section on Modular Networks and a background section
  on Subnetwork extractions. In my opinion, these can be merged.

## Minor changes

- The acronym ERM is used without introduction, but later specified in Section
  6. Either specify it at the start or not at all.

- In Table $4$ the methods are suddenly along the columns, while before they
  are along the rows. It would be more consistent to have the methods still as
  rows and the attributes along the columns.

**Strengths And Weaknesses:**

## Strengths:

- The proposed method does not rely on group-annotations, while still
  performing on par with methods that do need group-annotations.
- The authors motivate their method extensively and inform the reader about the
  thought process that went into developing all the components of their method.
  This was greatly appreciated!
- The authors perform clear experiments which show the usefulness of their
  method. The additional ablation analysis also further strengthens their
  results.
- The code that is provided is clearly written and allows for easy reproduction of the results.

## Weaknesses:

- The motivation that is given is mostly intuitive and although the empirical
  speak for themselves, a more theoretical justification would have been
  appreciated.
- A minor weakness is that the method relies on setting the number of clusters
  by hand. This is an additional hyperparameter and can probably be tuned using
  cross validation approach. An automated approach would be even better, but as this
  paper introduces a new approach I see this only as a minor weakness.

---

> ### Author Response · Authors · 2024-11-16
>
> We thank you for the careful review and thoughtful comments about our paper.
>
>
> ## Theoretical foundation
> We added a discussion and reference [1] (08/2024) to the end of Section 5.1. The authors in [1] investigate and provide a theoretical proof using a 2-layer fully connected neural network trained on the ColoredMNIST dataset (which exhibits a spurious correlation between each class and a specific color). They demonstrate that each group (defined by a label and its correlated spurious feature) can be separated early in training. Their theoretical proof on a simple network aligns with our empirical findings on more complex networks. We added reference [1] also to the related work and compared the results in Table 2.
>
>
> ## Choice of $k$
> A cross validation approach to determine $k$ would be detrimental to our method, as it requires knowledge about the spurious correlations and annotations of individual groups, whereas our method does not require any prior information about spurious correlations. However, the number of clusters is not a critical hyperparameter of our method as long as it is large enough. We added an analysis of $k$ in Section 7.2, which shows that the effectiveness of our approach is robust when $k \geq 8$.
>
>
> ## Changes
> We implemented all suggested changes, i.e., we
> - report mean and standard deviation for MGA and UAG
> - integrated the section on Modular Networks from related work to the background section on subnetwork extraction
> - specify ERM upon first usage
> - transposed  table 4 to have methods along the rows in order to be consistent throughout the paper
>
> [1]Yang et al., Identifying Spurious Biases Early in Training through the Lens of Simplicity Bias. AISTATS

---

> > ### Comment · Reviewer_54Z3 · 2024-11-26
> > **Response to Rebuttal**
> >
> > Thanks for providing a rebuttal and an adapted version of the paper. In particular the explanation and the analysis of the number of clusters is clear now and the added theoretical background is appreciated. Most of my concerns are thus addressed.

---

### Review · Reviewer_8JeD · 2024-10-22

**Summary Of Contributions:**

This paper tackles the problem of OOD generalization problem on datasets with spurious correlation. The major contributions are presented as follows:
- It is motivated by the cost of group annotations for spurious samples and authors propose an annotation-free framework to improve robustness to spurious correlation.
- They have two hypothesis: (a) masked subnet is more robust to spurious correlation, (b) ERM is prone to end up with a spurious-specific space than class-specific.
- Authors propose to extract a subnet for finetuning and also reduce spurious correlation by spurious-aware contrastive learning with clustering.
- Experiments show large improvement on multiple spurious correlations.

**Audience:**

Yes

**Claims And Evidence:**

Yes

**Requested Changes:**

Please refer to the weakness

**Strengths And Weaknesses:**

Pros:
- The paper is well-motivated, addressing the challenge of group annotation costs, and is well-organized.
- The experiments are comprehensive, covering multiple spurious correlations, and analyzing various hyper-parameters.
- The paper begins with clear hypotheses, which guide the reader logically to straightforward solutions.

Cons:

**Limited Novelty**
- My main concern lies in the limited novelty of the paper. The two hypotheses—(a) subnetworks are more robust to spurious correlations, and (b) spurious features tend to cluster more closely than class labels—are not entirely new.
- Hypothesis (a) has already been demonstrated by [1], while hypothesis (b) has been thoroughly explored in disentanglement, clustering, and domain-aware contrastive learning approaches (see [2,3,4,5]). Thus, the methodology largely builds on existing high-level intuitions from previous work.

**Missing clarification**
- There is a lack of clarity regarding the importance of sub-sampling. The authors introduce this concept in Section 5.2.2 without fully explaining its purpose. Several questions arise:
   1. Why not simply fine-tune on the original dataset $D$?
   2. For $D_{task}$ which contains hard cases, why is subnet masking/FT necessary? Could we not just fine-tune the ERM without subnet masking? This choice is not adequately justified, and no ablation experiments address it. Is this approach simply adopted from the debiasing idea in [1]?
- Table 5 raises additional concerns:
    1. I’m surprised that setting 5 performs worse than settings 6 and 7. Could the authors explain this discrepancy? Does the contrastive learning approach only work effectively when paired with subnet fine-tuning?
    2. Why does the vanilla subnet fine-tuning in settings 3, 6, and 7 perform significantly worse than full-model fine-tuning? This seems to contradict the assumptions in Figure 1.
    3. Why is there such a large difference in performance between training with and without $ConLoss_{FT}$  (settings 1 and 2)? How exactly does $ConLoss_{FT}$ differ from standard fine-tuning? The variation across the different combinations of modules seems excessive, making the results harder to interpret.



- [1] Training Debiased Subnetworks with Contrastive Weight Pruning
- [2] SelfReg: Self-supervised Contrastive Regularization for Domain Generalization
- [3] Domain Generalization via Contrastive Causal Learning
- [4] Feature Stylization and Domain-aware Contrastive Learning for Domain Generalization
- [5] Domain Generalization Using a Mixture of Multiple Latent Domains

---

> ### Author Response · Authors · 2024-11-16
>
> We thank you for the careful review and thoughtful comments about our paper.
> ## Novelty
> True, the two hypotheses are not entirely new. However, hypothesis (a) is not that subnetworks are *generally* more robust to spurious correlations, but that there *exists* a subnetwork that is more robust to spurious correlations. The crucial question is how to extract that subnetwork. The sub-sampling of minority (hard) cases is an integral part of our approach (see below). Your reference [1] (abbreviated as DCWP in our paper) builds on the same two hypotheses, yet our approach outperforms DCWP by a large margin (absolute gains in WGA between 16.1 and 30.0, comparable or better AVG).
>
>
>
>
> ## Clarification
> ### Purpose of Sub-Sampling
> Sub-Sampling is an integral part of our approach, to construct a balanced subset of the training data without spurious correlations. We use that sub-sampled dataset both to extract a subnetwork that is robust against spurious correlations and for fine-tuning, as fine-tuning on the original data is prone to just re-learn the spurious correlations We nevertheless fine-tuned on the original data after your question, but as expected, WGA dropped to 70.5 (compared to 89.6 when fine-tuning with the sub-sampled dataset, setting 2).
> We added a paragraph at the end of Section 5.2.2 to clarify the purpose of sub-sampling.
> ### Ablation Study - Table 5
> We revised Section 7.1 and redesigned Table 5 for easier readability to clarify the points.
>
> > For  which contains hard cases, why is subnet masking/FT necessary? Could we not just fine-tune the ERM without subnet masking? This choice is not adequately justified, and no ablation experiments address it. Is this approach simply adopted from the debiasing idea in [1]?
>
> Simply fine-tuning the whole network with the sampled data (including hard cases), but  without subnet masking is setting 6, which performs worse than first extracting a subnetwork (setting 2).
>
>
> > I’m surprised that setting 5 performs worse than settings 6 and 7. Could the authors explain this discrepancy? Does the contrastive learning approach only work effectively when paired with subnet fine-tuning?
>
> In all settings 5,6,7, we fine-tune with the balanced subset $D_{task}$. ERM models can effectively learn all features, but spurious features contribute more strongly to the prediction, due to the data distribution (DFR, Kirichenko et al. 2023). Hence, during fine-tuning, the model only has to adjust the classifier weights to rely more on invariant features. This is also confirmed by the increased WGA in setting 7 (fine-tuning only the last layer vs. fine-tuning all layers in setting 6). Adding the contrastive loss (setting 5), the model now has to optimize a dual objective: classification and representation, which is likely confusing rather than helpful, in particular as the balanced subset is comparably small. Similarly, adding the contrastive loss during fine-tuning after the extraction of a sub-network (setting 1) hurts performance. In this setting, the representation space has been optimized by the extraction of the subnetwork already and fine-tuning is only supposed to adjust the weight magnitudes after pruning. We hypothesize that forcing the network to also re-learn the representation space does not effectively align the remaining weights, but rather distorts weights and representations.
> We added the discussion to the last paragraph in Section 7.1.
>
> > Why does the vanilla subnet fine-tuning in settings 3, 6, and 7 perform significantly worse than full-model fine-tuning? This seems to contradict the assumptions in Figure 1.
>
> Setting 6 represents full model fine-tuning. In setting 3 we prune arbitrarily with sparse regularization, i.e., there is no criterion for improving the WGA. In setting 7, we only fine-tune the last layer with our balanced subset $D_{task}$, which does not distort the clusters induced by spurious features and thus, can not fully mitigate the reliance on spurious features.
>
> > Why is there such a large difference in performance between training with and without $ConLoss_{FT}$  (settings 1 and 2)? How exactly does $ConLoss_{FT}$ differ from standard fine-tuning?
>
> Standard fine-tuning only optimizes the classification loss, $ConLoss_{FT}$ in addition optimizes the representation space. We use $ConLoss_{FT}$ to extract a subnetwork with an appropriate representation space. For the performance difference, please see the second part of our answer on the question of setting 5 vs. 6 and 7.

---

> > ### Comment · Reviewer_8JeD · 2024-12-22
> >
> > Thanks for providing a rebuttal and an adapted version of the paper, which addressed part of my concerns.
> >
> > I am trying to organize the interesting contributions from the paper in my view while posing my concerns here.
> > - **Need to verify the effectiveness of pseudo-labeled group annotation in a clearer way.** IMO, in an ideal world, we could address the spurious correlation in OOD problem by having a debiased dataset at scale with no spurious correlation. Yet, it is hard to get such a dataset as the hardness of obtaining group annotations and lack of data. Therefore, the authors propose a new approach to identify the group annotations of samples by k-mean clustering. To understand the effectiveness of the proposed pseudo group annotation, it would be better to add some ablation showing this. Following are the example exps that might verify this point:
> >   - 1/ FT on whole dataset with the whole model
> >   - 2/ After 1/, post FT on the subset by GT group annotation with the whole model
> >   - 3/ After 1/, post FT on the subset by k-mean clustering with the whole model
> >   - 4/ After 1/, post FT on the subset by other pseudo-labeled group annotation (e.g, Bias-conflicting sample mining in DCWP) with the whole model.
> >
> >
> > - **Need to verify the effectiveness of Contrastive learning**. As you proposed in Figure 3, contrastive FT by leveraging group annotations (either GT or pseudo) is supposed to have more benefits than ERM.  When we only talk about the optimization loss in Table 5, setting 5-7, it seems that we cannot illustrate the effectiveness of contrastive FT with group annotations. Further, it seems like the proposed group annotation approach by clustering is only effective when coupled with Contrastive Pruning while hurt perf in other cases. This is kind of counter-intuitive to me. New paragraph on **Contrastive Loss during Fine-tuning** does not fully convince me here.

---

### Review · Reviewer_Syh8 · 2024-11-02

**Summary Of Contributions:**

This paper addresses the problem of neural networks being prone to relying on spurious correlations, such as when a neural network learns to distinguish images of water birds vs. land birds based on whether there is water or land in the background, rather than features of the birds themselves. Give a pre-trained dense neural network which has learned spurious associations, the authors propose a method to extract a subnetwork which relies less on the spurious associations to predict a label. A noteworthy feature of their method is that it does not require annotations with respect to spurious features, unlike many methods in the literature. In experiments, their method substantially outperforms other annotation-free methods and performs comparably to methods which require annotations.

As a caveat to this review, I have little expertise in the literature on reducing learning of spurious correlations in neural networks, so I am taking the authors at their word regarding the novelty of their method.

**Audience:**

Yes

**Claims And Evidence:**

Yes

**Requested Changes:**

- A more precise definition of spurious correlation, and more careful use of related language like spurious feature, spurious attribute, and shortcut.
- In section 4, the description of $\Pi$ doesn't quite make sense. The $\pi$ values are described as Bernoulli random variables, logits, and probabilities, which are all different.

**Strengths And Weaknesses:**

Strengths
=======
The method is clearly described and generally well motivated, and it makes intuitive sense. It is significant that it does not require annotating which instances possess the "spurious attribute," because as pointed out in the paper, the features that a neural network learns to rely on that do not generalize well may not be legible to humans, so annotation may not be feasible. The experiments are thorough and appropriate, and they show the proposed method strongly outperforming other annotation-free methods while performing comparably to methods that require annotation. Overall, this appears to be a useful method and a valuable contribution to the literature.


Weaknesses
==========
While the method is clear, the nature and scope of the problem it aims to address are not fully clear. Specifically, I'm not totally sure what the authors mean by "spurious correlation" or related phrases like "spurious attribute" and "spurious feature." In the abstract, a spurious correlation is defined as "features that have strong correlations with class labels but no causal relationship," while in the problem setting in section 3.1 it's defined as different groups having "different levels of bias correlated to an input feature of x and the label of y," which I don't understand.

Spurious correlation is a term that can potentially take on somewhat different meanings, and it's not clear what's intended here. For example, the motivating example in Figure 1 describes feature x1 (on the x-axis) as a spurious feature, while the correlation between being female and being blonde in the CelebA dataset is also described as spurious. But in CelebA, it's possible in principle to classify the blondness of images correctly without knowing whether the faces are male or female, whereas in Figure 1, x1 is not an optional feature; the other feature x2 is insufficient to separate the classes. So I'm not sure in what sense x1 is spurious, or whether it's the same sense in which being male/female is spurious in CelebA.

Overall, I found the Intro somewhat imprecise and muddled. I think it's probably more accessible to readers who are already familiar with the related literature, but it would be helpful to make clear up front what it meant by terms like "shortcuts" and what the groups are that are referred to in terms of group labels/annotations. These things only become somewhat clear later in the paper.

Relatedly, while I find Figure 3 very clear and helpful, I wonder when this scenario will and won't occur. For example, is it necessarily the case that decision boundaries cut cleanly in between clusters? Is it possible to have clusters in feature space along a spurious feature axis but have the decision boundary determined primarily by non-spurious features?

Overall, while I feel reasonably confident that the method is valuable, I think the paper would benefit from more precise discussion about the problem it aims to address. After all, the presence of water is pretty useful for identifying water birds vs. land birds, and many people would say that these things are causally related, even if the presence of water doesn't directly cause a bird to be a water bird.

One other major point: how is k chosen in the k-means clustering? Did I miss this somewhere?

---

> ### Author Response · Authors · 2024-11-16
>
> We thank you for the careful review and thoughtful comments about our paper.
>
>
> ## Terminology (spurious attributes/features/correlations, shortcuts)
>
> We adapted the beginning of the introduction to clarify the terms and streamlined their usage in the rest of the paper. In short:
>
> - _Spurious attributes/features_: Features that have no causal relationship with class labels (not necessarily correlated). Attributes merely refer to properties of the data / real world objects, whereas features can be abstractions thereof (potentially learned by the model). We try to make this distinction wherever appropriate, but otherwise use the two terms interchangeably. Spurious features are not harmful per se, as long as they are not correlated with a particular class.
>
> - _Spurious correlations_: Correlations of spurious features with class labels.
>
> - _Shortcuts_: Features that are easy to learn and sufficient to minimize the loss on the training data, but that do not generalize. They do not need to be spurious, but are not sufficient to fully describe the relationship between instances and class labels. For instance, the $x_1$ coordinate is a shortcut (sufficient to minimize loss on the training data, but will generalize poorly to unseen data as the other relevant feature $x_2$ is largely ignored), but not spurious (it has a causal relation). However, spurious features are often easy to learn and therefore prone to be used as shortcuts, in particular if they have a strong (spurious) correlation with class labels.
>
>
> ### Figure 1 - Terminology in example
> We revised Section 3 to clarify the distinction by the definitions provided above. Our goal is to provide a simple example that illustrates the following points: (i) there is an attribute ($x_1$) that is strongly correlated with the target label (y) during training; (ii) the machine learning model exploits this feature and underlearns the other feature ($x_2$); and (iii) our approach succeeds in forcing the model to rely on both attributes for the prediction.
>
> ## Prevalence of scenario sketched in Figure 3
> Indeed, the clusters in Figure 3, which describes the intuition or our approach, are necessarily simplified and idealized.  We agree that there may be cases where the decision boundary crosses through clusters, i.e., separates samples with the same spurious attribute. However, in our approach, we use clustering to define the contrastive batches for supervised contrastive learning. The supervised contrastive loss (eqn. 5) aims to bring samples of the same class closer together, which does not negatively affect the case where the decision boundary does not depend on the spurious features.The final loss (eqn. 7) includes both contrastive loss and cross-entropy loss.
>
> ## Choice of k for clustering
> We used $k=8$ in all our experiments and added this information to Section 5.3. We further added an analysis of $k$ in Section 7.2., observing  that the effectiveness of our approach is robust when $k \geq 8$. Hence, the number of clusters is not a critical hyperparameter for our approach, as long as it is large enough.
>
> ## Further requested changes
> We updated the description in Section 4 and dropped the notion of Bernoulli random variables to avoid confusion:  In our notation, $m_i$ is the binary mask, $\pi _i$ is a learned logit of a mask’s probability and $\sigma(\pi _i)$ the corresponding probability.

---

> > ### Comment · Reviewer_Syh8 · 2024-12-08
> >
> > I thank the authors for these changes. They have largely addressed my concerns, and I find the terminology and scope much clearer now.
> >
> > I still wish to quibble with this definition: "Spurious attributes or features are features that have no causal relationship with class labels." In causal inference, one important assumption which is used to relate probability distributions to causal DAGs is the Causal Markov condition, which gives rise to the slogan "no correlation without causation." That is, camels are correlated with the desert in images because they are connected in some kind of causal DAG, even if the presence of the desert in an image doesn't directly cause an animal to be a camel rather than a cow. So I believe that this could be phrased more precisely, maybe as "features for which there is no causal path between the feature and the class label."
> >
> > However, this is a relatively small concern, and it doesn't affect my recommendation or the overall strength of the paper.

---

### Decision · Action_Editor_gaW7 · 2025-01-07

**Recommendation:** Accept with minor revision

**Comment:**

This paper proposes PruSC, a novel method for mitigating the effect of spurious correlations in image classification without relying on group annotations. In response to reviewer feedback, the authors improved their definitions and terminology (e.g., of spurious correlation, feature, attribute, shortcut, etc.); clarified the purpose and importance of sub-sampling in the method; improved table 5; linked to some relevant theoretical work; conducted additional experiments and ablation studies. While some concerns remain, the paper addresses an important and clearly stated problem in machine learning which the reviewers believe is of interest to TMLR audience, and contains a thorough empirical evaluation of the proposed method with fairly strong results, thus warranting acceptance of this paper.

Below I copying reviewer comments. I highly recommend for the authors to see whether these additional experiments and edits can be added to the final version of the paper: (1) Consider adding experiments assessing the impact of pseudo-labeled group annotations (e.g., comparing the proposed approach with ground truth group annotations or alternative pseudo-labeling methods); (2) Provide more convincing evidence for the benefits of contrastive learning, particularly when combined with pseudo-group annotations. This could involve additional experiments or clearer analysis of existing results.

**Audience:**

The paper addresses an important problem of spurious correlations in machine learning, which is a significant challenge with implications for the robustness and generalization of models. The reviewers ultimately agreed that the paper will be of interest to the TMLR audience, and makes a valuable contribution to the field.

**Claims And Evidence:**

The claims made in the submission are supported by accurate, convincing, and clear evidence. The experiments are appropriate to demonstrate that the proposed method strongly outperforms other annotation-free methods while performing comparably to methods that require annotation.The experiments cover multiple spurious correlations and analyze various hyperparameters. The initial submission contained some poorly defined terminology and problem formulation, which has been improved during the rebuttal process based on reviewer feedback.